# The vulnerability assessment and obstacle factor analysis of urban agglomeration along the Yellow River in China from the perspective of production-living-ecological space

**Long Yang**[1], **Huihong Meng**[1], **Jitao Wang**[2]*, **Yifan Wu**[3], **Zhiwei Zhao**[1]

**1** School of Management, Zhengzhou University, Zhengzhou, China, **2** Henan Jintu Technology Group Co., LTD, Zhengzhou, China, **3** School of Economics, Xiamen University(Malaysia), Kuala Lumpur, Malaysia

* wangjitao521@hotmail.com

**Data Availability Statement:** All relevant data are within the manuscript.

**Funding:** This research was funded by [Key Project of Henan Province Higher Education (Humanities

## Abstract

Urban agglomerations are sophisticated territorial systems at the mature stage of city development that are concentrated areas of production and economic activity. Therefore, the study of vulnerability from the perspective of production-living-ecological space is crucial for the sustainable development of the Yellow River Basin and global urban agglomerations. The relationship between productivity, living conditions, and ecological spatial quality is fully considered in this research. By constructing a vulnerability evaluation index system based on the perspectives of production, ecology, and living space, and adopting the entropy value method, comprehensive vulnerability index model, and obstacle factor diagnostic model, the study comprehensively assesses the vulnerability of the urban agglomerations along the Yellow River from 2001 to 2020. The results reveal that the spatial differentiation characteristics of urban agglomeration vulnerability are significant. A clear three-level gradient distribution of high, medium, and low degrees is seen in the overall vulnerability; these correspond to the lower, middle, and upper reaches of the Yellow River Basin, respectively. The percentage of cities with higher and moderate levels of vulnerability did not vary from 2001 to 2020, while the percentage of cities with high levels of vulnerability did. The four dimensions of economic development, leisure and tourism, resource availability, and ecological pressure are the primary determinants of the urban agglomeration's vulnerability along the Yellow River. And the vulnerability factors of various urban agglomerations showed a significant evolutionary trend; the obstacle degree values have declined, and the importance of tourism and leisure functions has gradually increased. Based on the above conclusions, we propose several suggestions to enhance the quality of urban development along the Yellow River urban agglomeration. Including formulating a three-level development strategy, paying attention to ecological and environmental protection, developing domestic and foreign trade, and properly planning and managing the tourism industry.

and Social Sciences) grant number 21A170020]. The funder is Professor Zhao Zhiwei, who provided assistance with manuscript preparation, publication costs, etc.

**Competing interests:** The authors have declared that no competing interests exist.

## Introduction

Urban agglomerations are sophisticated territorial systems at the mature stage of city development that are concentrated areas of production and economic activity [1]. While enhancing the radiation-driven effect and implementing resource optimization configurations and energy supply, they can also promote the development of cities within the urban agglomeration. Therefore, observing and analyzing urban agglomerations can provide a more precise understanding of the dynamics of economic and social growth. At the sixth meeting of the Finance and Economy Commission of the CPC Central Committee in January 2020, General Secretary Xi Jinping underlined the need to promote high-quality development of central cities and urban agglomerations along the Yellow River [2]. "The Blue Book on the Development of the Yellow River Basin (2021)" suggests establishing an urban agglomeration in the Yellow River Basin as the core growth pole. However, com-pared to other regions, the ecological environment and economic and social development conditions are relatively unequally distributed among the urban agglomerations along the Yellow River. The development of towns and cities has resulted in issues such as irrational spatial layout and energy structure, which increase the vulnerability of the urban agglomerations [3, 4].

As the main territorial units of urban agglomerations, cities are characterized by significant vulnerability. Prior to becoming a crucial instrument in the analysis of global environmental change and sustainable development [5], vulnerability studies were initially used to examine the characteristics of disasters and poverty [6]. At present, the majority of academic research on vulnerability is based on provincial [7], municipal [8], regional [9, 10], and other scales. Studies in this field explore the definition [11], analytical framework [12], comprehensive measurement and evaluation [13], and influencing factors [14] of vulnerability. The research perspectives focus mainly on disaster risk [15], the composite human-earth system [16], climate [17–19], energy [20, 21], socio-ecology [22], and so on. Urban vulnerability research has become a crucial scientific method for exploring the harmonious symbiosis of human and environmental systems [13]. However, most of the current studies are based on the quantity and scale of cities and lack a comprehensive assessment from the perspective of the quality of urban development. Therefore, we aim to develop a comprehensive evaluation system for urban vulnerability to serve as a reference for assessing the overall quality of urban development and promoting sustainable development for both people and the environment. In recent years, with the gradual popularization of the "three pillars" concept in the international community, academics have increasingly focused on the study of territorial space from a production-living-ecological perspective [23, 24]. The integrated spatial layout of production-living-ecological space plays a crucial role in harmonizing the relationship between people and the land and provides a methodology for improving the quality of urban development and reducing urban vulnerability. At present, the research on production-living-ecological space mainly focuses on two aspects. On the one hand, a production-living-ecological space evaluation index system is developed. As an illustration, Yang et al used geospatial data and social statistics to establish a system of indicators for evaluating the production-living-ecological functions of rural areas in the Beijing-Tianjin-Hebei region and to measure the coordination of the production-living-ecological functions in the rural areas in the Beijing-Tianjin-Hebei region [25]. On the other hand, it is founded on the optimization of production-living-ecological space. For instance, Yang et al conducted a land use layout optimization and multi-scenario study of the Lanzhou-Xining urban agglomeration in China based on a production-living-ecological spatial perspective and explored the future development of urban space in the agglomeration [26]. Through the literature review, we found that existing studies have involved the Yellow River Basin, but lacked vulnerability analysis of urban agglomerations along the Yellow River.

Due to the special characteristics of the natural environment and geographic location, the human-land relationship in the Yellow River Basin shows dynamic changes, and the urban crisis has existed for a long time. Therefore, the study of vulnerability from the perspective of production-living-ecological space is crucial for the sustainable development of the Yellow River Basin and global urban agglomerations.

Taking seven urban agglomerations along the Yellow River as the research area, this article constructed a vulnerability measurement index system based on the perspective of production-living-ecological space quality, and comprehensively evaluated the vulnerability of urban agglomerations along the Yellow River from 2001 to 2020. Through the measurement results, we analyzed the overall vulnerability and internal spatial differences of the urban agglomeration, and used the barrier factor diagnostic model to identify the influencing factors of barrier factors and their evolution trends.

The innovations and contributions of this article are: (1) Taking the seven urban agglomerations along the Yellow River as the research object, a better understanding of the integrity and systematicity of the Yellow River urban agglomerations will help promote integration and coordination among cities. The research results have certain reference significance for urban management planning and sustainable development in other similar regions around the world. (2) On the basis of existing research, we establish an index system for evaluating vulnerability that is more comprehensive and efficiently evaluates the vulnerability of the urban agglomerations along the Yellow River between 2001 and 2020. From the perspective of production-living-ecological space, the original assessment index system is synthesized, and new indicators are creatively introduced in conjunction with the real conditions of the research region. (3) We analyze the vulnerability assessment results from the perspective of the overall vulnerability and spatial variability of the urban agglomeration. Furthermore, the obstacle factor model was used to identify the influencing factors, and then the evolution trend of the obstacle factors was analyzed. Based on these, targeted management measures are proposed for the urban agglomeration along the Yellow River.

The following is the arrangement of the paper's other sections: The Materials section describes the study area and data sources. The Methods section presents the methodology, including the vulnerability assessment model, vulnerability measurement index model, and obstacle factor diagnostic model. TheResults section explains the results and includes overall vulnerability analysis, variability analysis within urban agglomerations, and barrier factor analysis. The Discussion section discusses the results. The Conclusions section provides the conclusions and outlook of the paper.

## Materials

### Study area

The Yellow River is the birthplace of Chinese civilization and an important ecological barrier and economic zone in China. As a spatial carrier to promote the construction of ecological civilization, the Yellow River Basin has a prominent position in the overall layout of China's economic development, energy production, social stability and opening-up. According to "the National Master Plan for Functional Zone" and related urban agglomeration plans, the research objects of this paper are seven urban agglomerations through the Yellow River Basin, which are Lanxi urban agglomeration, Ningxia urban agglomeration, Hubao and Eyu urban agglomeration, Guanzhong Plain urban agglomeration, Jinzhong urban agglomeration, Central Plains urban agglomeration and Shandong Peninsula urban agglomeration. The specific cities are shown in Table 1.

**Table 1. Specific overview of the urban agglomerations along the Yellow River.**

| Urban agglomeration | Specific city areas |
|---|---|
| Lanxi urban agglomeration | Lanzhou, Baiyin, Dingxi, Xining and Haidong, |
| Ningxia urban agglomeration | Yinchuan, Shizuishan, Wuzhong, Zhongwei |
| Hubao and Eyu urban agglomeration | Hohhot, Baotou, Ordos, Yulin |
| Guanzhong Plain urban agglomeration | Yuncheng, Linfen, Xi'an, Baoji, Xianyang, Tongchuan, Weinan, Shangluo, Tianshui, Pingliang, Qingyang. |
| Jinzhong urban agglomeration | Taiyuan, Jinzhong, Xinzhou, Luliang, Yangquan |
| Central Plains urban agglomeration | Zhengzhou, Kaifeng, Luoyang, Pingdingshan, Xinxiang, Jiaozuo, Xuchang, Luohe, Jiyuan, Hebi, Shangqiu, Zhoukou, Jincheng |
| Shandong Peninsula urban agglomeration | Yantai, Weihai, Jinan, Weifang, Linyi, Rizhao, Zaozhuang, Heze, Jining, Liaocheng, Binzhou, Dongying, Dezhou, Qingdao, Tai'an and Zibo. |

Note: According to the "Development Plan for the Central Plains Urban Agglomeration," "Development Plan for the Guanzhong Plain Urban Agglomeration," "Development Plan for the Shandong Peninsula Urban Agglomeration 2021–2035," "Development Plan for the Hubei, Inner Mongolia, and Shanxi Urban Agglomerations," "Development Plan for the Lanzhou-Xining Urban Agglomeration," "New Urbanization "14th Five-Year Plan" for the Ningxia Hui Autonomous Region," and "Development Plan for the Central Shanxi Urban Agglomeration", etc.

The population of the seven urban agglomerations in the Yellow River Basin accounts for approximately 78% of the total basin population. Moreover, the GDP generated by these urban agglomerations represents about 60% of the basin's overall GDP [27]. These statistics highlight the vital role urban that agglomerations play in supporting the economic growth, energy production and ecological preservation of the Yellow River Ba-sin. However, the urban agglomerations along the Yellow River exhibit significant disparities in ecological conditions and levels of socioeconomic development, leading to noticeable spatial imbalances in high-quality development. This situation is accompanied by prominent issues such as low economic quality, unbalanced energy structure, and severe environmental challenges. The specific mani-festations are as follows:

1. Poor ecological environment: the urban agglomerations within the Yellow River Basin are characterized by detrimental ecological conditions. According to the Ministry of Ecology and Environment's 2021 Chinese ambient air quality rating, 12 out of the 20 cities with the worst air pollution can be found in this region.

2. Low vitality of resident consumption: this is another challenge in the urban ag-glomera-tions along the Yellow River. According to the "China Urban Living Circle Vitality Index 2021," only two provincial capitals, Zhengzhou and Xi'an, ranked among the top 20 in terms of the comprehensive score of the urban living circle vitality index within the Yellow River Basin urban agglomeration.

Given these challenges, it is crucial to conduct a comprehensive analysis of the vulnerabil-ities faced by the urban agglomerations along the Yellow River. This analysis should take into account the production, living, and ecological spaces within the region. Furthermore, region-specific development proposals should be formulated to narrow regional disparities, optimize the energy environment, promote ecological protection, and facilitate high-quality development in the Yellow River Basin. The lessons learned from these initiatives can then be applied to enhance the sustainable development of other urban agglomerations around the world.

## Data sources

This article focuses on assessing the vulnerability of seven urban agglomerations in the Yellow River Basin from 2001 to 2020. To conduct this assessment, the researchers selected 9 first-level indicators and 27 second-level indicators. These indicators were used to analyze the vulnerability factors and their evolution trends through a barrier factor diagnostic model. The data for the indicators related to agricultural production, social security services, ecological pressure, and ecological governance were obtained from "the Statistical Yearbooks" of each province in the Yellow River Basin. The data for indicators related to non-agricultural production, economic development, living services, and resource supply were sourced from "the China Urban Statistical Yearbook". The data at the tourism and leisure functions level came from "the Statistical Bulletins" of various cities. For the indicators related to per capita water resources at the resource supply level, the data came from "the Water Resources Bulletin". The carbon emission data per 10,000yuan of GDP were obtained from Chen's research [28]. In cases where some data were missing, interpolation methods were used to supplement the missing values.

## Methods

### Vulnerability assessment model

**Index system construction.**    From the perspective of production-living-ecological space, Thie article builds a comprehensive evaluation index system based on the concepts of representativeness, comprehensiveness, scientificity, rationality, and operability, considering the findings of previous research. Every detail of the Yellow River urban agglomeration's actual situation is taken into account. Three views are used in the construction of the comprehensive assessment index system: the production space quality, the living space quality, and the ecological space quality. In terms of the selection of production space quality and living space quality indicators, Zhang et al. 's research on production-life-ecological function of urban agglomeration in the middle reaches of the Yangtze River is mainly referred to [29]. For the selection of indicators for ecological space quality, Fang et al.'s research on the spatial differentiation of urban fragility in China is primarily consulted [13]. Although existing research has established a vulnerability evaluation index system, there are still some shortcomings that need to be addressed. For instance, existing indicators lack comprehensiveness when measuring agricultural production, as they only evaluate the overall output of agriculture without considering agricultural machinery power. Similarly, when measuring non-agricultural production, they only take industrial production into account, while ignoring the tertiary industry. Moreover, indicators for measuring the quality of living services only consider economic income and basic security, without taking into account urban population pressure, a crucial factor affecting citizens' lives. Lastly, the indicators for measuring the quality of living space do not keep up with the times, as they do not include the important impact of tourism and leisure on residents' quality of life.

Therefore, this study proposes four improvements. Firstly, to more accurately evaluate agricultural production capacity, new indicators are added, such as Per unit of land gross power of agricultural machinery. Secondly, to fully assess non-agricultural production capability, per unit of land output of secondary and tertiary industries are included. Thirdly, a new dimension related to tourism and leisure is added in the measurement of living space quality. This is measured using indicators such as per unit of land domestic tourist arrivals and per unit of land domestic tourism revenue, providing a more accurate representation of residents' quality of life in the Yellow River urban agglomeration. Finally, minor adjustments are made to the

wording of indicators for ecological space quality based on the specific characteristics of the Yellow River urban agglomeration.

In conclusion, a vulnerability assessment index system for the Yellow River urban agglomeration was constructed, including 9 first-level indicators and 27 second-level indicators (as shown in Table 2).

**Entropy method.** The entropy method is a widely used weight allocation method in multi-criteria decision making, aimed at determining the relative importance of each index or criterion in the decision-making process. This method evaluates the importance of each factor by assessing the entropy of each index based on the concept of information entropy and

**Table 2. Vulnerability evaluation indicator system for urban agglomerations along the Yellow River.**

| Goal layer | Subsystem | Classification Layer | Indicator Layer | Indicator Attributes | Weights |
|---|---|---|---|---|---|
| Assessment of Urban Vulnerability in the Yellow River Urban Agglomerations | The quality of the production space | agricultural production | X1 Per unit of land grain yield | + | 0.1028 |
| | | | X2 Per unit of land gross power of agricultural machinery | + | 0.0996 |
| | | | X3 Per unit of land gross agricultural, forestry, fishery and livestock production | + | 0.1035 |
| | | non-agricultural production | X4 Per unit of land gross industrial product | + | 0.0976 |
| | | | X5 Per unit of land output of secondary and tertiary industries | + | 0.1202 |
| | | economic development | X6 Per unit of land investment in fixed assets | + | 0.1110 |
| | | | X7 Economic density | + | 0.1275 |
| | | | X8 Actual utilization of foreign capital | + | 0.1153 |
| | | | X9 Total exports and imports | + | 0.1225 |
| | The quality of the living space | living services | X10 Population density | + | 0.0991 |
| | | | X11 GDP per capita | + | 0.1087 |
| | | | X12 Per capita road area | + | 0.1054 |
| | | | X13 Per capita total retail sales of consumer goods | + | 0.1093 |
| | | social security services | X14 Number of health facility beds per 10,000 population | + | 0.0993 |
| | | | X15 Internet penetration | + | 0.0807 |
| | | | X16 Education expenditure ratio | + | 0.0809 |
| | | | X17 Ratio of urban to rural incomes | + | 0.0955 |
| | | tourism and leisure functions | X18 Per unit of land domestic tourist arrivals | + | 0.1074 |
| | | | X19 Per unit of land domestic tourism revenue | + | 0.1137 |
| | The quality of the ecological space | resource supply | X20 Per capita water resources | + | 0.1345 |
| | | | X21 Per capita green space | + | 0.1463 |
| | | ecological pressure | X22 Energy consumption of 10,000yuan of GDP | - | 0.1257 |
| | | | X23 Industrial wastewater discharges per 10,000yuan of GDP | - | 0.1196 |
| | | | X24 Carbon emissions per 10,000yuan of GDP | - | 0.1204 |
| | | ecological governance | X25 Greening coverage in built-up areas | + | 0.1164 |
| | | | X26 Non-hazardous domestic waste disposal rate | + | 0.1103 |
| | | | X27 Comprehensive industrial solid waste utilization rate | + | 0.1268 |

assigns the corresponding weight. Compared to methods such as the Delphi method, expert survey method, and analytic hierarchy process, the entropy method excludes subjective factors in the weighting process, resulting in more objective index weights [30]. This eliminates the influence of varying data, allowing for the comparison of the evaluation of objects using a unified standard over the years. To calculate and evaluate the vulnerability of urban agglomerations along the Yellow River, this paper first utilizes the range method to standardize the data for each index, then applies the entropy method. The specific calculation process follows:

Firstly, data preprocessing, such as data standardization, should be conducted. Prior to applying the entropy method for weight assignment, it is essential to address the irregularities in data direction and the significant differences in data magnitudes and units. This can be achieved through forward or reverse data processing. These are the calculable equations:

Forward processing:

$$X_{ij} = \frac{X_{ij} - \min X_{ij}}{\max X_{ij} - \min X_{ij}} \tag{1}$$

Reverse processing:

$$X_{ij} = \frac{\max X_{ij} - X_{ij}}{\max X_{ij} - \min X_{ij}} \tag{2}$$

Where $x_{ij}$ represents the original data of j indicator in $i$ year, $X_{ij}$ is the standardized value, $max$-$x_{ij}$ is the highest value of j indicator in $i$ year, and $minx_{ij}$ is the minimum value of j indicator in $i$ year.

Next, apply the entropy method to assign weights. The entropy method utilizes the concept of information entropy to measure the uncertainty in each indicator. Higher entropy values indicate higher uncertainty and less information, while lower entropy values signify lower uncertainty and more information. By considering the degree of variation in each indicator and the information carried by the entropy method, the information entropy becomes a valuable tool for calculating the weights of each indicator. These weights form the basis for a comprehensive evaluation of multiple indicators. These are the calculable equations:

$$Y_{ij} = \frac{X_{ij}}{\sum_{i=1}^{n} X_{ij}} \tag{3}$$

$$e_j = -\frac{1}{\ln n} \sum_{i=1}^{n} Y_{ij} \ln Y_{ij} \tag{4}$$

$$w_j = \frac{1 - e_j}{\sum_{j=1}^{m} 1 - e_j} \tag{5}$$

Where $Y_{ij}$ represents the dimensionless value, $e_j$ represents the entropy value of the $j$th indicator, and $w_j$ represents the weight of the $j$th indicator.

**Vulnerability measurement index model.**   After multiplying the normalized values obtained from Eqs (1) and (2) with the weights calculated from Eq (5), the vulnerability of the urban agglomeration along the Yellow River from 2001 to 2020 can be derived through weighted summation. The vulnerability can be classified by referring to relevant literature

**Table 3. Classification of vulnerability.**

| Vulnerability index | [0,0.2] | [0.2,0.3] | [0.3,0.4] | [0.4,0.5] | ≥0.5 |
|---|---|---|---|---|---|
| Vulnerability level | Lower vulnerability | Low vulnerability | Moderate vulnerability | High vulnerability | Higher vulnerability |
| Representational state | Excellent state | Good state | Average state | Alert state | Dangerous state |

[31]. (as shown in Table 3). The calculation equation is:

$$UVAI = \sum_{i=1}^{m} w_j X_{ij} \tag{6}$$

## Obstacle factor diagnostic model

The purpose of vulnerability assessment of urban agglomerations is not only to evaluate the development level of each city in the region, but more importantly to find the obstacle factors affecting urban vulnerability, so as to adjust the current urban development behaviors and policies. In order to further enhance the urban agglomeration's development status and de-crease its vulnerability, this paper uses the obstacle factor diagnosis model [32] to pin-point the effects of different dimensions and indicators on the urban agglomeration's vulnerability and identify the primary development-restraining obstacles. Three primary indices utilized to assess and diagnose the obstacle factor diagnosis model are the factor contribution degree, index deviation degree, and obstacle degree. The index deviation degree indicates the distance between each indication and the ideal goal, the obstacle degree reflects the effect of a single indicator on the system, and the factor contribution degree measures the contribution of a single component to the overall goal. The calculation equations are.

1. The factor contribution degree:

$$F_{ij} = w_i \times p_{ij} \tag{7}$$

   Where $F_{ij}$ is the factor contribution degree, $w_i$ is the weight of the $i$th subsystem, and $p_{ij}$ is the weight of the $j$th indicator of the ith subsystem.

2. The index deviation degree:

$$V_i = 1 - X_{ij} \tag{8}$$

   Where $V_i$ is index deviation degree, and $X_{ij}$ is the standardized value of a single indicator.

3. The obstacle degree:

$$C_{ij} = \frac{F_{ij} \times V_{ij}}{\sum_{i=1}^{n} F_{ij} \times V_{ij}} \times 100\% \tag{9}$$

   Where $C_{ij}$ is obstacle degree, and the larger the value, the higher the degree of obstacle that the indicator has for the vulnerability of the urban agglomeration along the Yellow River.

## Results

Based on the production-living-ecological space perspective, this article measures the vulnerability and obstacle factors of the urban agglomeration in the Yellow River Basin from 2001 to

2020. Subsequently, we conducted analyzes from three aspects: overall vulnerability, internal differences in urban agglomerations, and key obstacle factors and their evolution.

## Overall vulnerability analysis

The vulnerability of the seven urban agglomerations along the Yellow River from 2001 to 2020 is expressed graphically, as shown in Fig 1.

As can be seen from Fig 1, the vulnerability levels of the urban agglomeration along the Yellow River are as follows: Shandong Peninsula urban agglomeration>Central Plains urban agglomeration > Hubao and Eyu urban agglomeration >Jinzhong urban agglomeration >Guanzhong Plain urban agglomeration >Lanxi urban agglomeration > Ningxia agglomeration urban agglomeration along the Yellow River. Overall, the spatial differentiation of the vulnerability of urban agglomerations is characterized significantly. The overall vulnerability shows an obvious three-level gradient distribution of high, medium and low degrees, corresponding to the lower reaches, middle reaches and upstream areas of the Yellow River Basin respectively. Specifically, first of all, the urban agglomerations located at the first level of vulnerability gradient are the Shandong Peninsula urban agglomeration and the Central Plains urban agglomeration. From 2001 to 2020, they continuously scored higher than 0.4 on the vulnerability index, indicating a high level of vulnerability. The reason is that both of them are China's most populous and energy-rich provinces. Although this ensures their strong economic development, social security and resource supply capabilities, the process of rapid development will inevitably lead to massive energy consumption and massive discharge of waste. These exceed the carrying capacity of the environment and cause greater ecological pressure, resulting in a much higher vulnerability than other urban agglomerations. Therefore, easing the pressure on per capita energy consumption and optimizing the diversified energy structure are particularly important for the Shandong Peninsula urban agglomeration and Central Plains urban agglomeration in Shandong.

Secondly, the urban agglomeration located at the second level gradient (moderate) of vulnerability is the Hubao and Eyu urban agglomeration. During the study period, its

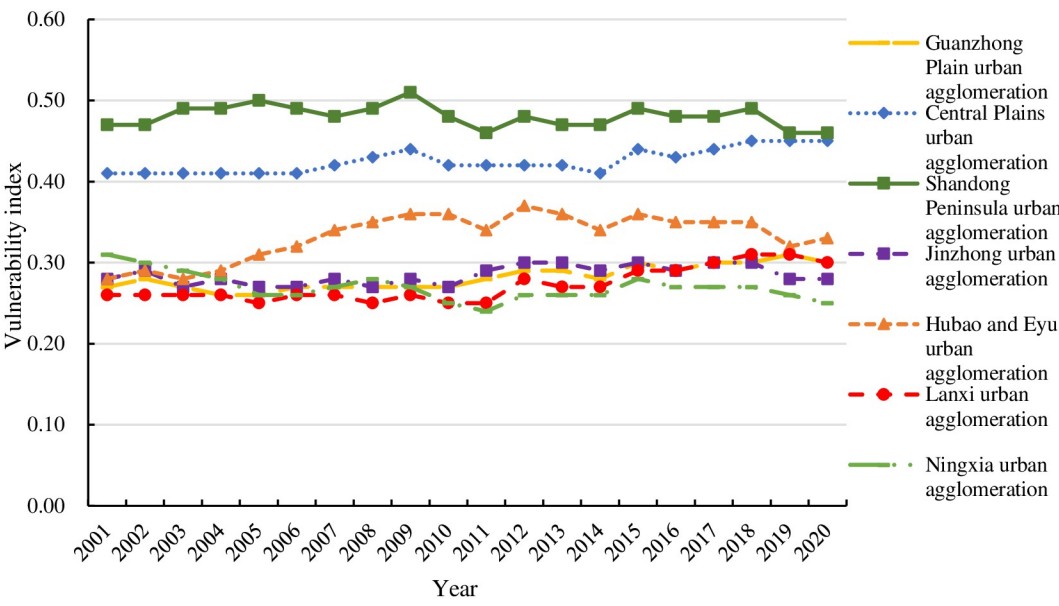

**Fig 1. Urban agglomerations along the Yellow River's vulnerability trends, 2001–2020.**

vulnerability index generally increased from 0.28 in 2001 to 0.33 in 2020, and the vulnerability level increased from low vulnerability to moderate vulnerability. The reason is that the economic development level of these cities is relatively backward and the infrastructure construction is incomplete. As a result, the radiating and driving capabilities of the central cities are not strong, and the level of co-construction and sharing of public services needs to be improved urgently. Since resource sharing and mutual benefit between the central city and surrounding cities are encouraged, the Hubao and Eyu urban agglomeration should develop sensible urban planning and enhance infrastructure construction. This will hasten the process of regional coordinated development.

Thirdly, the third echelon (low) of vulnerability includes Jinzhong urban agglomeration, Guanzhong Plain urban agglomeration, Lanxi urban agglomeration, and Ningxia urban agglomeration, all of which are relatively low vulnerability Among them, the vulnerability of Jinzhong urban agglomeration, Guanzhong Plain urban agglomeration, and Lanxi urban agglomeration is relatively stable, while the Ningxia urban agglomeration has the greatest degree of change. Its vulnerability index dropped from 0.31 in 2001 to 0.25 in 2020, and its vulnerability level dropped from medium vulnerability to low vulnerability. Limited by geographical area and resource conditions, in the early stages of development, the economic development of Ningxia urban agglomeration was slow, the comprehensive economic level was not high, and the index of vulnerability was high. However, due to the region's abundant land resources and good agricultural basic conditions, coupled with the vigorous development of tourism in the later period, which has promoted local agricultural production, economic development and resource protection, the vulnerability has been declining year by year.

## Difference analysis within urban agglomerations

In order to further explore the characteristics of spatial differentiation within the urban agglomeration along the Yellow River, we integrated the ArcGIS 10.2 software and calculated the average vulnerability of each city within the urban agglomeration for four different time periods: 2001–2005, 2006–2010, 2011–2015, and 2016–2020. The results were then visually expressed (as shown in Table 4).

Overall, the percentage of cities with medium and higher vulnerability levels has remained constant, while the proportion of highly vulnerable cities has been on the rise, and the central cities within each urban agglomeration consistently face a high degree of vulnerability. This trend can be attributed mainly to the rapid economic growth experienced by these cities, with greater energy consumption, which has led to increased ecological and environmental burdens during the initial stages of development. As a result, the ecological carrying capacity of these cities in the later stages of development has been strained. Subsequently, we conduct specific analyzes on the vulnerability of urban agglomerations in different time periods:

(1) 2001–2005

During this period, the percentage of cities with medium vulnerability was 29.31%, while the percentages of cities with high and higher vulnerability were 24.14% and 12.07%, respectively. The Shandong Peninsula urban agglomeration exhibited high vulnerability, primarily in cities such as Qingdao, Yantai, Weihai, Jinan, and Zibo. During the study period, these cities had larger economic densities, higher secondary industrial output values per unit of land, higher industrial output values per unit of land, and more pressure on energy consumption. Moreover, lower expenditure on education further con-tributed to their vulnerability. In the Central Plains urban agglomeration, Zhengzhou experienced the highest level of vulnerability, followed by Kaifeng and other cities. This can be attributed to Zhengzhou's high population density, limited availability of water re-sources per unit of land, and inadequate development

**Table 4. Analysis of spatial variability within the urban agglomerations along the Yellow River, 2001–2005, 2006–2010, 2011–2015 and 2016–2020.**

| Urban agglomeration | City | Vulnerability level | | | |
|---|---|---|---|---|---|
| | | 2001–2005 | 2006–2010 | 2011–2015 | 2016–2020 |
| Lanxin urban agglomeration | Lanzhou | 3 | 3 | 3 | 3 |
| | Baiyin | 1 | 1 | 2 | 2 |
| | Dingxi | 2 | 2 | 2 | 2 |
| | Xining | 3 | 3 | 3 | 3 |
| | Haidong | 2 | 2 | 2 | 2 |
| Ningxia urban agglomeration | Yinchuan | 5 | 3 | 3 | 3 |
| | Shizuishan | 3 | 3 | 3 | 3 |
| | Wuzhong | 1 | 1 | 1 | 2 |
| | Zhongwei | 1 | 1 | 1 | 1 |
| Hubao and Eyu urban agglomeration | Hohhot | 3 | 3 | 3 | 3 |
| | Baotou | 3 | 3 | 3 | 3 |
| | Ordos | 2 | 3 | 5 | 3 |
| | Yulin | 2 | 2 | 2 | 2 |
| Guanzhong Plain urban agglomeration | Yuncheng | 2 | 2 | 2 | 2 |
| | Linfen | 2 | 2 | 2 | 1 |
| | Xi'an | 4 | 4 | 4 | 4 |
| | Baoji | 2 | 2 | 3 | 33 |
| | Xianyang | 3 | 3 | 3 | 3 |
| | Tongchuan | 2 | 2 | 2 | 2 |
| | Weinan | 2 | 2 | 2 | 3 |
| | Shangluo | 2 | 2 | 2 | 2 |
| | Tianshui | 2 | 2 | 2 | 2 |
| | Pingliang | 2 | 2 | 2 | 2 |
| | Qingyang. | 1 | 1 | 1 | 2 |
| Jinzhong urban agglomeration | Taiyuan | 5 | 5 | 5 | 5 |
| | Jinzhong | 5 | 5 | 5 | 5 |
| | Xinzhou | 2 | 1 | 2 | 2 |
| | Lvliang | 1 | 2 | 2 | 1 |
| | Yangquan | 3 | 3 | 3 | 3 |
| Central Plains urban agglomeration | Zhengzhou | 4 | 4 | 4 | 4 |
| | Kaifeng | 5 | 5 | 5 | 5 |
| | Luoyang | 3 | 3 | 3 | 5 |
| | Pingdingshan | 3 | 3 | 3 | 3 |
| | Xinxiang | 3 | 5 | 3 | 5 |
| | Jiaozuo | 5 | 5 | 5 | 5 |
| | Xuchang | 5 | 5 | 5 | 5 |
| | Luohe | 5 | 5 | 5 | 5 |
| | Jiyuan | 3 | 3 | 3 | 3 |
| | Hebi | 5 | 5 | 5 | 5 |
| | Shangqiu | 3 | 3 | 3 | 3 |
| | Zhoukou | 3 | 3 | 3 | 5 |
| | Jincheng | 3 | 3 | 3 | 3 |

(*Continued*)

**Table 4.** (Continued)

| Urban agglomeration | City | Vulnerability level | | | |
|---|---|---|---|---|---|
| | | 2001–2005 | 2006–2010 | 2011–2015 | 2016–2020 |
| Shandong Peninsula urban agglomeration | Yantai | 4 | 4 | 4 | 4 |
| | Weihai | 4 | 4 | 4 | 4 |
| | Jinan | 4 | 4 | 4 | 4 |
| | Weifang | 5 | 5 | 5 | 5 |
| | Linyi | 3 | 3 | 3 | 3 |
| | Rizhao | 5 | 5 | 5 | 5 |
| | Zaozhuang | 5 | 5 | 5 | 5 |
| | Heze | 3 | 3 | 3 | 3 |
| | Jining | 5 | 5 | 5 | 5 |
| | Liaocheng | 5 | 5 | 5 | 5 |
| | Binzhou | 3 | 3 | 3 | 3 |
| | Dongying | 4 | 5 | 4 | 4 |
| | Dezhou | 3 | 5 | 5 | 5 |
| | Qingdao | 4 | 5 | 4 | 4 |
| | Tai'an | 5 | 5 | 5 | 5 |
| | Zibo | 4 | 4 | 4 | 4 |

Note: "1" represents lower vulnerability, "2" represents low vulnerability, "3"represents moderate vulnerability, "4"represents high vulnerability, and "5" represents higher vulnerability.

of mechanized agricultural production during the early stages of the study. Additionally, as one of the oldest cities in China, Kaifeng has a large number of ancient buildings and ruins due to its unique geographical location and historical reasons. This resulted in early city planning having to consider the protection of cultural relics and avoid artificial damage. As a result of the damage, Kaifeng's urban infrastructure construction has been affected to a certain extent, and various expenditures and costs have increased significantly, making it more vulnerable.

(2) 2006–2010

In this phase, the proportion of cities with medium vulnerability is 29.31%, while those with high vulnerability and higher vulnerability account for 24.14% and 13.79%, respectively. Within the Shandong Peninsula urban agglomeration, Dezhou and Dongying both have upgraded their vulnerability levels to higher. This can be attributed to significant changes in land use structure caused by rapid economic development during the study period, resulting in a decrease in the city's carrying capacity and an in-crease in vulnerability. In the Ningxia urban agglomeration, Yinchuan has experienced a decrease in vulnerability from higher to moderate. The main reason is that the construction of the urban economic belt along the Yellow River was proposed in 2005. This policy was carried out with Yinchuan as the center, which brought a lot of policy preferences and resource support to the development of Yinchuan. Yinchuan has invested a lot of investment in urban construction and built an urban transportation network in accordance with the concept of "near connections, distant connections, and external expansion". At the same time, Yinchuan actively implements the strategy of moving the city center westward, accelerating the development of new areas and the renovation of old cities. In addition, a large number of landscaping and wetland protection projects have been implemented to promote the improvement of the urban ecological environment and reduce the vulnerability of Yinchuan.

(3) 2011–2015

During this period, the proportion of cities with medium vulnerability was 34.48%, while those with high vulnerability and higher vulnerability accounted for 25.86% and 12.07%, respectively. Within the Jinzhong urban agglomeration, Xinzhou's vulnerability level increased from lower to low. This change was primarily influenced by the decline in the coal energy market in 2013, which resulted in increased downward pressure on the economy. Additionally, the initiation of the Green for Grain Project also contributed to the change in vulnerability, which has increased the pressure on agricultural production. In the Hubao and Eyu urban agglomeration, Erdos's vulnerability level increased from moderate to higher. This was mainly due to the city's rapid industrial development during this period. The per unit of land gross industrial product significantly increased from 420,000 yuan per square kilometer in 2006 to 2,460,000 yuan per square kilometer in 2015. However, this growth was accompanied by a decrease in the comprehensive utilization rate of solid industrial waste from 70% in 2006 to 42% in 2015, which increased the city's vulnerability. This shows that Ordos has not taken into account the ecological environment while rapidly developing its economy. It is an extensive development at the expense of the environment, which is not conducive to the sustainable development of the city. But on the other hand, there were also cities whose vulnerability declined during this period, namely Xinxiang in the Central Plains urban agglomeration, which dropped from high vulnerability to moderate vulnerability. At the first annual meeting of the Low Carbon China Forum in 2010, Xinxiang was awarded the title of "Actively Developing Low-Carbon Economic City", the only city in the Central Plains urban agglomeration to be awarded the title. Following that, Xinxiang made a concerted effort to create a low-carbon circular economy and placed a high value on technological advancement and industrial innovation. Simultaneously, it employs multifaceted spatial planning techniques to enhance the low-carbon spatial arrangement, mitigate urban susceptibility, and promote the city's sustainable development.

(4) 2016–2020

During this period, the proportion of cities with medium vulnerability was 29.31%, while those with high vulnerability and higher vulnerability accounted for 29.31% and 12.07%, respectively. Ordos in the Hubao and Eyu urban agglomeration, Lvliang in the Jinzhong urban agglomeration, and Linfen in the Guanzhong Plain urban agglomeration have decreasing vulnerability in this stage. The period from 2016 to 2020 is China's "Thirteenth Five-Year Plan" period. It is a stage when China's economic development changes from high-speed development to high-quality development. Urban development shifted from solely focusing on economic growth in terms of speed and scale to emphasizing coordinated development across the economy, society, and environment. During the study period, these cities vigorously improved resource utilization efficiency, built environmentally friendly industries, and focused on the maintenance and restoration of the ecological environment, which improved the quality of urban tertiary space and reduced urban vulnerability. Moreover, due to the impact of the COVID-19 epidemic in 2020, human activities have decreased, and the pressure on infrastructure and the environment has decreased. This results in ecological restoration and reduced urban vulnerability.

## Obstacle factor analysis

To calculate the obstacle degree of each indicator to urban vulnerability of urban ag-glomerations along the Yellow River, Eqs (7)–(9) were utilized. However, due to the large number of indicator factors involved in the evaluation index system, only the top ten influencing factors are filtered as the main obstacle factors for in-depth discussion. Additionally, the data between 2001 and 2020 were chosen as samples for obstacle factor analysis due to the large sample size

of the data from 2001 to 2020, which mainly includes key obstacle factor analysis and obstacle factor evolution analysis. (as shown in Table 5).

**Key obstacle factor analysis.** Table 5 shows that four dimensions of economic development, leisure and tourism, resource availability, and ecological pressure are the primary determinants of the urban agglomeration's vulnerability along the Yellow River. Among these dimensions, the availability of green space for human habitation is identified as the primary limiting factor, followed by the total volume of imports and exports. Focusing on the key vulnerability factors in each urban agglomeration, we identify the shortcomings of the vulnerability of each urban agglomeration and propose corresponding precise governance measures to improve the city's carrying capacity and reduce their vulnerability. The following approaches can be adopted:

Firstly, targeting the Shandong Peninsula urban agglomeration and Central Plains urban agglomeration at the first level of vulnerability gradient. The most critical factors causing vulnerability are per capita water resources and per capita green area, which are the resource and supply factors of ecological space quality indicators. With the continuous expansion of cities, the pressure on resources and the environment brought by economic development will

**Table 5. Obstacle factors and obstacle degree in 2001 and 2020 for the urban ag-glomerations along the Yellow River.**

| Urban Agglomeration | Year | Item | 1 | 2 | 3 | 4 | 5 | 6 | 7 | 8 | 9 | 10 |
|---|---|---|---|---|---|---|---|---|---|---|---|---|
| Lanxi urban agglomeration | 2001 | A | X21 | X9 | X7 | X5 | X8 | X19 | X22 | X3 | X18 | X1 |
| | | B | 5.46 | 5.06 | 4.93 | 4.77 | 4.76 | 4.45 | 4.34 | 4.28 | 4.17 | 4.08 |
| | 2020 | A | X21 | X9 | X7 | X8 | X5 | X19 | X6 | X18 | X22 | X1 |
| | | B | 5.70 | 5.48 | 5.46 | 5.17 | 5.15 | 4.93 | 4.65 | 4.54 | 4.52 | 4.50 |
| Ningxia's urban agglomeration | 2001 | A | X9 | X20 | X8 | X7 | X19 | X3 | X18 | X5 | X6 | X2 |
| | | B | 5.93 | 5.86 | 5.60 | 5.60 | 5.12 | 4.99 | 4.56 | 4.47 | 4.46 | 4.23 |
| | 2020 | A | X20 | X9 | X7 | X8 | X5 | X19 | X6 | X18 | X27 | X1 |
| | | B | 5.92 | 5.55 | 5.53 | 5.23 | 5.22 | 5.11 | 4.80 | 4.78 | 4.52 | 4.40 |
| Hubao and Eyu urban agglomeration | 2001 | A | X7 | X9 | X5 | X8 | X6 | X23 | X21 | X3 | X1 | X18 |
| | | B | 5.34 | 5.20 | 5.03 | 4.94 | 4.63 | 4.57 | 4.57 | 4.49 | 4.41 | 4.41 |
| | 2020 | A | X7 | X9 | X5 | X19 | X6 | X18 | X23 | X8 | X1 | X3 |
| | | B | 5.69 | 5.66 | 5.36 | 5.18 | 5.03 | 4.94 | 4.92 | 4.81 | 4.61 | 4.60 |
| Jinzhong urban agglomeration | 2001 | A | X21 | X9 | X8 | X20 | X23 | X7 | X3 | X5 | X19 | X1 |
| | | B | 5.66 | 5.21 | 4.99 | 4.88 | 4.88 | 4.76 | 4.52 | 4.44 | 4.41 | 4.33 |
| | 2020 | A | X21 | X9 | X8 | X7 | X20 | X5 | X19 | X23 | X3 | X1 |
| | | B | 5.39 | 5.20 | 5.03 | 5.02 | 4.88 | 4.72 | 4.68 | 4.56 | 4.35 | 4.35 |
| Shandong Peninsula urban agglomeration | 2001 | A | X21 | X24 | X22 | X20 | X9 | X23 | X8 | X3 | X11 | X17 |
| | | B | 6.59 | 6.00 | 5.92 | 5.61 | 5.58 | 5.51 | 4.98 | 4.43 | 4.00 | 3.86 |
| | 2020 | A | X20 | X21 | X24 | X19 | X8 | X22 | X9 | X18 | X6 | X23 |
| | | B | 6.44 | 6.40 | 5.45 | 5.41 | 5.28 | 5.21 | 5.13 | 5.00 | 4.92 | 4.82 |
| Central Plains urban agglomeration | 2001 | A | X21 | X9 | X20 | X8 | X24 | X22 | X11 | X23 | X12 | X7 |
| | | B | 6.57 | 5.96 | 5.57 | 5.56 | 5.27 | 4.87 | 4.59 | 4.41 | 3.93 | 3.92 |
| | 2020 | A | X21 | X20 | X8 | X9 | X22 | X24 | X23 | X7 | X12 | X19 |
| | | B | 6.57 | 6.55 | 6.07 | 5.77 | 5.38 | 5.27 | 4.72 | 4.56 | 4.54 | 4.52 |
| Guanzhong Plain urban agglomeration | 2001 | A | X21 | X9 | X8 | X7 | X5 | X20 | X3 | X23 | X11 | X24 |
| | | B | 5.72 | 4.95 | 4.81 | 4.69 | 4.50 | 4.26 | 4.24 | 4.23 | 4.15 | 4.13 |
| | 2020 | A | X21 | X9 | X7 | X5 | X19 | X20 | X8 | X23 | X18 | X6 |
| | | B | 5.82 | 5.26 | 5.25 | 4.98 | 4.76 | 4.68 | 4.62 | 4.52 | 4.29 | 4.23 |

Note: A represents the obstacle factor and B represents the obstacle degree.

become more obvious. The difference is that the vulnerability of the Shandong Peninsula urban agglomeration to carbon emissions per 10,000 yuan of GDP is significantly higher than that of the Central Plains urban agglomeration, which is another important factor restricting urban development. Although Qingdao is the second batch of low-carbon pilot cities in my country, its carbon emissions per 10,000 yuan of GDP have dropped by 27%. However, other cities in the Shandong Peninsula urban agglomeration, such as Binzhou, have three times the carbon emissions per 10,000 yuan of GDP than Qingdao. The pressure on energy consumption is greater, exacerbating urban vulnerability. On the other hand, the Central Plains urban agglomeration's total import and export volume and actual utilization of foreign capital are significantly more vulnerable than the Shandong Peninsula urban agglomeration, which are two other important factors restricting its development. Combined with the geographical location of the Central Plains urban agglomeration not facing the sea or border, as well as the unique hydrological and geological conditions of the Yellow River, the shipping value is very low. The cities in the Central Plains urban agglomeration face difficulties in import and export trade and are hindered in attracting investment, resulting in both the total city import and export volume and the actual amount of utilized foreign investment being low. Therefore, it is advisable for the Shandong Peninsula urban agglomeration to prioritize ecological restoration projects and focus on developing a robust green recycling economy and diversified energy structure. While the Central Plains urban agglomeration should fully utilize the location advantages of the central region, implement the recycling economy development model, strengthen the expansion of the scale of foreign trade, and unleash the "speed of the Central Plains".

Secondly, for the Hubao and Eyu urban agglomeration, which is in the second echelon of vulnerability. The obstacles affecting its development mainly focus on the economic development dimension and non-agricultural production dimension. The main reason is that the local industrial structure. During the study period, the Hubao and Eyu urban agglomeration formed an industrial system dominated by energy, chemical industry, metallurgy, etc. However, the added value of urban secondary and tertiary industries in this region accounts for a lower proportion of GDP than the national average. This proves that its industrial structure is unreasonable and relies too much on industrial economic development, resulting in a single industrial structure. At the same time, its technological innovation is insufficient and its energy utilization efficiency is low, resulting in greater pressure on economic development and higher urban vulnerability. Therefore, the Hubao and Eyu urban agglomeration should strengthen investment in science and technology and the introduction of talents to promote the upgrading of production technology. And we must also promote the optimization of industrial structure and develop diversified industries.

Thirdly, for the Ningxia urban agglomeration, Jinzhong urban agglomeration, Guanzhong Plain urban agglomeration and Lanxi urban agglomeration, which are at the third level of vulnerability gradient. Its vulnerability factors mainly focus on economic development, tourism and leisure functions and resource supply dimensions. Specifically, most of these cities are located in the inland northwest of China, have weak economic foundations and little international trade, and lack the ability to attract foreign investment. Even if they relied on abundant natural resources and tourism resources, developing tourism has promoted economic development. The large values, high proportions and high rankings of the obstacle factors in the tourism and leisure function and resource supply dimensions indicate that the economic returns of the tourism industry in these urban agglomerations are low, and resource consumption and environmental sacrifices are large, which has had a negative effect on urban vulnerability. Therefore, for these urban agglomerations in the third echelon of vulnerability, local governments should focus on improving resource utilization efficiency and promoting industrial

transformation and upgrading. At the same time, they should deepen opening up internally and externally, and promote regional cooperation and industrial transformation.

**Obstacle factor evolution analysis.**   In order to gain a deeper understanding of the evolution trend of the vulnerability of urban agglomerations along the Yellow River, we next conducted the following comparative analysis of the obstacle factors of each urban agglomeration in 2001 and 2002.

Firstly, for the urban agglomeration of the first echelon of vulnerability. During the study period, the ranking of energy consumption per 10,000yuan of GDP of the Shandong Peninsula urban agglomeration dropped from third to sixth, and the obstacle degree dropped from 5.92 to 5.21. Similarly, the factor ranking of industrial wastewater discharge of 10,000yuan of GDP decreased from the sixth to the tenth, and its obstacle degree decreased from 5.51 to 4.82. It can be seen that the Shandong Peninsula urban agglomeration has achieved certain results in energy structure adjustment and environmental governance. However, there is insufficient supply of clean energy, and there is a high dependence on high-carbon energy sources, leading to high carbon emissions per 10,000yuan of GDP. Regarding the Central Plains urban agglomeration, the disappearance of the obstacle factor per capita GDP indicates that during the study period, the overall economic development of the Central Plains urban agglomeration was strong, and productivity significantly improved. Furthermore, it should be mentioned that in 2020, both the Shandong Peninsula urban agglomeration and the Central Plains urban agglomeration saw an increase in the obstacle factors, i.e., per unit of land domestic tourist arrivals and per unit of land domestic tourism revenue, which are the key indicators that we focused on in this study. This observation implies that in the course of the development and evolution of urban agglomerations, the focus of development has shifted from primary and secondary industries towards the tertiary industry. The rapid development of the tertiary industry, represented by tourism, has put significant pressure on urban transportation, energy consumption, and the ecological environment. As a result, the vulnerable first-tier urban agglomerations should tighten up on planning and control in the tourism industry. The development of low-carbon tourist projects and goods, as well as the establishment of a strong tourism management system and efficient resource management, are imperative. By reducing energy consumption in the tourism industry, these urban agglomerations can promote sustainable urban development.

Secondly, for the urban agglomeration of the second echelon of vulnerability. During the study period, obstacle factors related to the tourism and leisure function dimensions were added to both the Hubao and Eyu urban agglomeration in 2020. These included per unit of land domestic tourist arrivals and per unit of land domestic tourism revenue, which came in at number four and sixth, respectively, with 5.19 and 4.94 obstacle degrees. Other obstacle factors did not change significantly. This shows that with the development of urban agglomerations, the significance of tourism and leisure functions is growing. From a specific perspective, the number of domestic tourists and domestic tourism income in Erdos and Yulin, within the Hubao Eyu City cluster, are relatively low. There is also limited tourism exchange between these cities. As a result, the overall level of tourism development in these cities is low, which makes it difficult to meet the tourism and leisure needs of local residents and optimize the urban industrial structure. Therefore, for the second level of vulnerable city clusters, it is essential to strengthen the optimization and upgrading of the industrial structure. This can be achieved by promoting the development of the tourism industry and facilitating tourism exchange and collaboration between cities. These efforts will contribute to the transformation and development of the city clusters, providing them with the necessary impetus for growth.

Thirdly, regarding the third echelon of vulnerable urban agglomeration s, including the Ningxia urban agglomeration, the Jinzhong urban agglomeration, the Guanzhong Plain urban

agglomeration, and the Lanxi urban agglomeration, the ranking of obstacle factors has remained relatively stable compared to the previous two echelons. These urban agglomerations are generally more stable. From the perspective of obstacle degree, it can be divided into two types for analysis: The first type includes Ningxia urban agglomeration and Jinzhong urban agglomeration along the Yellow River. The vulnerability levels of production, living, and ecological spaces in these two city clusters have mostly decreased. This indicates that, with the Ningxia urban agglomeration as the main focus, efforts to promote coordinated development of the Yellow River Ecological Economic Belt and the Northern Green Development Zone have achieved certain results. The quality of urban construction and development has improved, and the regional spatial pattern continues to be optimized. On the contrary, the other types are Guanzhong Plain urban agglomeration and Lanxi urban agglomeration, and the obstacle degree of each factor in them is mostly increased. It can be seen that although these cities have lower vulnerability, they still face different challenges in the development process. Development planning needs to be tailored to local conditions, and regional industrial cooperation needs to be further improved.

## Discussion

Based on the above research results, we compare this article with previous research results to illustrate the scientific nature and marginal contribution of this article.

Overall, the urban agglomeration along the Yellow River has a remarkable three-level gradient differential in terms of vulnerability. This characteristic is the same as earlier research, and closely corresponds to the overall vulnerability gradient differentiation of Chinese urban agglomerations [13]. This demonstrates that, as a whole, the degree of vulnerability of the urban agglomeration along the Yellow River examined in this research is congruent with the real circumstances in China. Because the gradient of vulnerability at all levels is generally consistent with previous research on urban areas in the upper, middle, and lower ranges of the Yellow River, it suggests that the study is scientific. On the other hand, the total vulnerability of the urban agglomeration along the Yellow River indicates an increasing trend, based on the specific value of the obstacle degree. The percentage of cities with high susceptibility rises, while the percentage of cities with moderate or high vulnerability stays constant. This is consistent with previous research conclusions on urban agglomerations in the Yangtze River Basin [33, 34], Pearl River Basin [35] and other regions, but is contrary to research conclusions in northeast China [36], central Yunnan [37], Zhangjiakou [38] and other regions of China. It is evident that the vulnerability of the urban agglomeration along the Yellow River has a particular influencing mechanism.

Urban agglomeration is a coupled system of economy, society and environment. Prior research on the Yellow River Basin has mostly concentrated on environmental and economic issues [39–42]. The findings indicate that the primary determinants of the urban agglomeration's growth along the Yellow River are resource availability, ecological environment, and economic development. The research conclusions of this article have a high degree of fit with it. However, existing studies lack the perspective of living space quality. Although there are studies on social and people's livelihood aspects, they are more focused on policy support and lack research on urban tourism and leisure functions [43, 44]. It can be seen that the existing research on the quality of urban development in the Yellow River Basin is incomplete. Therefore, this paper constructs a multi-dimensional index system to comprehensively evaluate the evolutionary characteristics of vulnerability barrier factors in the urban agglomeration along the Yellow River. Compared with existing evaluation methods, this index system is more systematic and scientific.

Contrary to earlier research, the study's findings indicate that the influence of leisure and tourism activities on the vulnerability of urban agglomerations along the Yellow River has grown dramatically and is especially significant. We proceed from an economic perspective based on supply and demand effects. In terms of demand, with the continuous development of China's economy and the continuous increase in national income, the demand for tourism is becoming increasingly strong. This has led to a continuous increase in the number of tourists per region, which has put forward higher-level requirements for the development of tourism in urban agglomerations along the Yellow River; At the same time, as China's economy shifts from high-speed development to high-quality development, the country has attached great importance to the protection of the Yellow River culture and the development of the tourism industry since 2020. Higher requirements have been placed on the average tourism income of urban agglomerations along the Yellow River, which has led to the increasingly prominent importance of tourism and leisure functions in the development of urban vulnerability. In terms of supply, urban agglomerations along the Yellow River mainly focused on the development of primary and tertiary industries in the past, with the tertiary industry accounting for a relatively low proportion. The tourism supply was seriously insufficient and difficult to meet social needs. Although the industrial structure has improved in recent years and the tertiary industry has developed, the development level of the tourism industry is not high, mostly at the expense of consuming resources and damaging the environment. The average number of tourists per area and the average tourism income are not high, which is extremely detrimental to the high-quality development of the city and increases the vulnerability of the urban agglomeration along the Yellow River.

In summary, the marginal contributions of this article are as follows: This article improves the existing indicator system and conducts a scientific analysis of the overall vulnerability and obstacle factor evolution of the urban agglomeration along the Yellow River. The high-quality development level of urban agglomerations along the Yellow River is studied not only from economic and environmental aspects, but also from social aspects. It was found that the tertiary industry, specifically tourism and leisure functions, is particularly important to urban vulnerability. Theoretically, this article enriches previous research on the Yellow River Basin and provides a new perspective on tourism and leisure research. In practice, the research results can guide the planning and management of urban agglomerations along the Yellow River, and provide reference for their sustainable development.

## Conclusions

By constructing a vulnerability assessment index system and utilizing the entropy value method, the vulnerability comprehensive measure index model, and the obstacle factor diagnostic model in conjunction with the spatial analysis function of Arcgis10.2, we measure the vulnerability of the urban agglomeration along the Yellow River, analyze the spatial variability within the urban agglomeration from 2001 to 2020, and analyze the influencing factors and evolution trends of obstacle factors. Based on these, the following conclusions are drawn, which can provide valuable lessons for sustainable urban development, both in China and globally:

1. The spatial differentiation characteristics of urban agglomeration vulnerability are significant. A clear three-level gradient distribution of high, medium, and low degrees is seen in the overall vulnerability; these correspond to the lower, middle, and upper reaches of the Yellow River Basin, respectively. The specific vulnerability levels are: Shandong Peninsula urban agglomeration > Central Plains urban agglomeration > Hubao and Eyu urban ag-

glomeration > Jinzhong urban agglomeration > Guanzhong Plain urban agglomeration > Lanxi urban agglomeration > Ningxia urban agglomeration.

2. The percentage of cities with higher and moderate levels of vulnerability did not vary from 2001 to 2020, while the percentage of cities with high levels of vulnerability did. The central cities of each urban agglomeration are always highly vulnerable, and the surrounding cities are slightly less vulnerable than them.

3. There are differences in the vulnerability barrier factors of each urban agglomeration along the Yellow River. The four dimensions of economic development, leisure and tourism, resource availability, and ecological pressure are the primary determinants of the urban agglomeration's vulnerability along the Yellow River.

4. From 2001 to 2020, the vulnerability factors of various urban agglomerations showed a significant evolutionary trend, the obstacle degree values have declined, and the importance of tourism and leisure functions has gradually increased.

Based on the above conclusions, we propose several suggestions to enhance the quality of urban development along the Yellow River urban agglomeration.

1. Formulate a three-level development strategy. Based on the vulnerability of each urban agglomeration, local governments should formulate urban agglomeration development plans based on local conditions. Highly vulnerable urban agglomerations should implement a green circular economy model, moderately vulnerable cities need to focus on optimizing industrial structure and upgrading production technology, and low-vulnerability urban agglomerations should accelerate opening up and industrial regional cooperation. In particular, the urban agglomeration of Shandong Peninsula ought to initiate ecological restoration initiatives and cultivate a varied energy framework. The creation of a circular economy model and bolstering the growth of international commerce scale should be the priorities of the Central Plains urban agglomeration. The metropolitan agglomerations of Yu, Baotou, and Hubei should increase their investments in research and technology, introduce more professional talents, and support the modernization of production methods and the optimization of the industrial structure. The urban agglomerations of Jinzhong, Guanzhong Plain, Lanxi, and Ningxia along the Yellow River are expected to foster coordinated development and strengthen regional collaboration.

2. Pay attention to ecological and environmental protection. Local governments should coordinate and reconcile the contradiction between ecology and development and adhere to sustainable development. For central cities with high vulnerability, some functions of the city should be distributed to surrounding cities to alleviate the pressure on the urban ecological environment; while surrounding cities should strengthen urban greening construction and promote regional cooperation to form an urban ecosystem and realize inter-city collaborative development. At the same time, local governments should establish ecological restoration projects. For upstream urban agglomerations such as Ningxia urban agglomeration, the construction of green ecological corridors should be accelerated to ensure the integrity and stability of the ecosystem in the Yellow River Basin. For urban agglomerations along the lower reaches of the Yellow River such as the Central Plains urban agglomeration, it is necessary to Continue to promote pollution prevention and control, carbon reduction and emission reduction, and enhance the ecological barrier function of the Yellow River Basin.

3. Develop domestic and foreign trade. Due to locational constraints, the level of opening-up of urban agglomerations along the Yellow River is relatively effective. In this regard,

governments at all levels should increase infrastructure investment and construction, establish a high-quality transportation network throughout the Yellow River Basin, and promote urban interconnection. At the same time, we should take Shandong Peninsula as the window of regional foreign trade, and vigorously develop export-oriented industries and carry out import and export business. And provide preferential policies to attract regional investment, improve the level of foreign trade development.

4. Properly plan and manage the tourism industry. Regional governments should give full play to the leading role of the tourism industry. Based on the regional Yellow River cultural advantages, relevant policies have been introduced to develop tourism into a regional advantageous industry. Urban agglomerations with high tourism and leisure vulnerability should optimize tourism resource management, enhance tourism resource utilization efficiency, and develop low-carbon tourism products and projects to reduce energy consumption in the tourism industry. For urban agglomerations with a low level of tourism industry development, they should promote technological upgrading and structural optimization and upgrading of the tourism industry by accelerating the introduction of tourism professionals and promoting talent training and information exchange. At the same time, each urban agglomeration should also vigorously promote urban tourism cooperation. This is the main way to promote the complementary advantages of the tourism industry in the upper, middle and lower reaches of the Yellow River Basin and narrow regional differences. The ultimate goal is to promote coordinated regional development.

Of course, we must admit that this study has some shortcomings. Firstly, due to limitations in data availability and processing methods, this study only calculated a few factors reflecting tourism and leisure functions. More indicators can be added to future research for more comprehensive measurements. Secondly, although we have conducted a more in-depth analysis of the vulnerability and vulnerability factors of the urban agglomeration along the Yellow River, there is a lack of further discussion on the spatial impact relationship of each urban agglomeration. In future research, we can learn from more related spatial econometrics techniques to conduct more in-depth horizontal spatial research on the urban agglomeration along the Yellow River, thereby providing more reference suggestions for the coordinated development of the urban agglomeration. Finally, the urban vulnerability effect has complex feedback mechanisms and hysteresis effects, which will bring certain uncertainties to the results and require more follow-up research.

## Acknowledgments

The authors wish to thank the peer reviewers for improving this manuscript.

## Author Contributions

**Conceptualization:** Huihong Meng, Jitao Wang.

**Data curation:** Huihong Meng.

**Formal analysis:** Jitao Wang.

**Funding acquisition:** Jitao Wang, Zhiwei Zhao.

**Investigation:** Yifan Wu.

**Methodology:** Long Yang, Huihong Meng.

**Project administration:** Jitao Wang.

**Resources:** Huihong Meng, Zhiwei Zhao.

**Software:** Huihong Meng, Jitao Wang.

**Supervision:** Yifan Wu.

**Validation:** Long Yang, Yifan Wu, Zhiwei Zhao.

**Visualization:** Huihong Meng.

**Writing – original draft:** Huihong Meng.

**Writing – review & editing:** Long Yang.

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
