## [Decision Letter · Decision Letter 0]

7 Nov 2023

PONE-D-23-34145The vulnerability assessment and obstacle factor analysis of urban agglomeration along the Yellow River in China from the perspective of production-living-ecological spacePLOS ONE

Dear Dr. Yang,

Thank you for submitting your manuscript to PLOS ONE. After careful consideration, we feel that it has merit but does not fully meet PLOS ONE’s publication criteria as it currently stands. Therefore, we invite you to submit a revised version of the manuscript that addresses the points raised during the review process.

Reviewer 1 Comments:

Specific suggestions are as follows：

1. The structure of the abstract is incomplete. The author adds the suggestions or measures proposed and relevant data to revise the abstract.

2. Line 56 to 58, this paper says “However, compared to other regions, the ecological environment and economic……. urban agglomerations along the Yellow River”, Which articles did the author read to reach this conclusion, need to be explained, or use relevant data to illustrate this view.

3. Line 68 to 73, this paper summarizes the previous research, which is conducted from the quantitative scale, and evaluates it from the quality scale. I have read the full text, from your use of methods and selected indicators, it does not seem to reflect from perspective of quality. In the part of introduction, you can add content of research model.

4. In the part of literature review, the innovation of this paper and the contribution of previous studies have not been clearly expressed. It is suggested to refer to the following:

[1] The local coupling and telecoupling of urbanization and ecological environment quality based on multisource remote sensing data. Journal of Environmental Management.

5. In the part of data sources, be specific about where your data comes from, the data of each indicator comes from relevant documents to be clearly explained.

6. In the Methodology, whether there are evidences for the selection of all indicators, which need to explain. Why is the entropy method used to calculate the weight in this article? Are other methods of calculating index weight not feasible?

7. Line 164 to 218, why are there two identical subheadings?

8. Line 232 to 236, the explanation of three formulas is not clear, the reader may not understand it. You can write “where is…..”.

9. Line 310 to 347, the paper analyzes the rise of vulnerability from 2006 to 2010, 2011 to 2015 and 2016 to 2020. Therefore, the vulnerability of some urban agglomerations has decreased during these periods, should it also be specified?

10. Line 368 to 420, The proposed approaches or suggestions should be formulated in conjunction with the conclusion, making targeted approaches or suggestions rather than general approaches or suggestions. This paper only puts forward the measures of each urban agglomeration, authors can add the approaches of different cities in each urban agglomeration.

11. The discussion is not deep enough. No targeted policy recommendations are put forward for the research conclusions of this paper. It is suggested to refer to the following:

[2] Local and tele-coupling development between carbon emission and ecologic environment quality. Journal of Cleaner Production.

12. Some references, words and grammar have some problems in the format of the article. For example：

(1) in the keywords: barrier facto?

(2) Formula (2) is incorrectly in the format of article.

(3) Line 320, “led to increased government support” revises “led to increase ……”.

(4) Line 348, “3.3.Barrier factor analysis” differs from other subheadings title above.

(5) Line 367, “The following approach can be adopted”, you can write “The following approaches can be adopted”

(6) In the references. The 20th reference check please.

(7) Fig.2. need to be adjusted. Y-axis name and Y-axis scale need to be adjusted. The English font needs to be adjusted in the Fig.

and so on. The format of article can refer to the journal requirements.

Reviewer 2 Comments:

The authors analyze the vulnerability and obstacle factors of seven urban agglomerations along the Yellow River from the perspective of production-living-ecological space. Although the topic is practically meaningful, the manuscript lacks innovation and academic contribution. Firstly, examining urban vulnerability from the perspective of production-living-ecological space is not an original research idea, as numerous existing studies have adopted this approach. Secondly, the research methods are quite singular and superficial, relying mainly on basic techniques like entropy method and weighted summation to determine indicator weights and vulnerability indices, without employing more sophisticated and innovative assessment methods. Thirdly, the empirical analysis is overly simple and descriptive, without delving deeper into the inherent mechanisms driving the vulnerability of each urban agglomeration. Finally, the manuscript lacks targeted policy recommendations that provide clear guidance on how to reduce the vulnerability of urban agglomerations. In summary, the innovation and academic contribution of this study are inadequate. I would encourage the authors to further enhance the originality and depth of the research to increase its academic value. Extensive revisions and supplements are needed for the paper to meet the requirements for publication. Specific recommendations are as follows:

1 Introduction

(1)Line 92: The introduction lacks a comprehensive review of the existing literature in the field. It is essential to provide a clear articulation of how this study differentiates itself from prior research. The introduction should conclude with a paragraph that emphasizes the novel contributions and innovations of this research, thereby underscoring its significance in the academic discourse.

2 Materials and Methods

(1)Line 174: The choice of the proposed vulnerability measurement and barrier factor diagnosis models seems arbitrary. A more robust theoretical foundation is required to justify the selection of these models over potential alternatives. This would enhance the rationale and ensure that the methodology is grounded in established economic theories.

(2)Line 183: The introduction of the entropy method is cursory. A detailed exposition of its core concepts, strengths, and potential limitations is necessary. This will not only justify its selection for evaluating vulnerability indicators but also provide readers with a better understanding of its applicability and relevance.

3 Results

(1)Line 355: The barrier factor analysis, while identifying key dimensions, falls short in explaining the underlying mechanisms that influence vulnerability. A more in-depth discussion, rooted in economic theory, is imperative to shed light on the significance and interplay of these dimensions. This will also help in contextualizing the results within the broader economic framework.

4 Discussion

(1)Line 427: The discussion on policy implications is rather generic. Given the diverse barrier factors faced by different urban agglomerations, it is crucial to propose more nuanced and actionable recommendations tailored to the specific challenges of each agglomeration. This would make the findings more relevant for policymakers and practitioners.

(2)Line 465: The conclusion should acknowledge limitations and gaps of the current study, and discuss future improvements.

Reviewer 3 Comments:

1. The reasons for the selected indicators in the indicator system should be explained;

2. The discussion section should clearly indicate the marginal contribution of this article and compare it with existing research results;

3. Language expression needs improvement.

We look forward to receiving your revised manuscript.

Kind regards,

Fuyou Guo, (Ph.D.

Academic Editor

PLOS ONE

Journal Requirements:

2. We note that Figures 1 & 3 in your submission contain [map/satellite] images which may be copyrighted. All PLOS content is published under the Creative Commons Attribution License (CC BY 4.0), which means that the manuscript, images, and Supporting Information files will be freely available online, and any third party is permitted to access, download, copy, distribute, and use these materials in any way, even commercially, with proper attribution. For these reasons, we cannot publish previously copyrighted maps or satellite images created using proprietary data, such as Google software (Google Maps, Street View, and Earth). For more information, see our copyright guidelines: http://journals.plos.org/plosone/s/licenses-and-copyright.

a. You may seek permission from the original copyright holder of Figures 1 & 3 to publish the content specifically under the CC BY 4.0 license. 

“I request permission for the open-access journal PLOS ONE to publish XXX under the Creative Commons Attribution License (CCAL) CC BY 4.0 (http://creativecommons.org/licenses/by/4.0/). Please be aware that this license allows unrestricted use and distribution, even commercially, by third parties. Please reply and provide explicit written permission to publish XXX under a CC BY license and complete the attached form.

Additional Editor Comments (if provided):

Reviewer 1 Comments:

Specific suggestions are as follows：

1. The structure of the abstract is incomplete. The author adds the suggestions or measures proposed and relevant data to revise the abstract.

2. Line 56 to 58, this paper says “However, compared to other regions, the ecological environment and economic……. urban agglomerations along the Yellow River”, Which articles did the author read to reach this conclusion, need to be explained, or use relevant data to illustrate this view.

3. Line 68 to 73, this paper summarizes the previous research, which is conducted from the quantitative scale, and evaluates it from the quality scale. I have read the full text, from your use of methods and selected indicators, it does not seem to reflect from perspective of quality. In the part of introduction, you can add content of research model.

4. In the part of literature review, the innovation of this paper and the contribution of previous studies have not been clearly expressed. It is suggested to refer to the following:

[1] The local coupling and telecoupling of urbanization and ecological environment quality based on multisource remote sensing data. Journal of Environmental Management.

5. In the part of data sources, be specific about where your data comes from, the data of each indicator comes from relevant documents to be clearly explained.

6. In the Methodology, whether there are evidences for the selection of all indicators, which need to explain. Why is the entropy method used to calculate the weight in this article? Are other methods of calculating index weight not feasible?

7. Line 164 to 218, why are there two identical subheadings?

8. Line 232 to 236, the explanation of three formulas is not clear, the reader may not understand it. You can write “where is…..”.

9. Line 310 to 347, the paper analyzes the rise of vulnerability from 2006 to 2010, 2011 to 2015 and 2016 to 2020. Therefore, the vulnerability of some urban agglomerations has decreased during these periods, should it also be specified?

10. Line 368 to 420, The proposed approaches or suggestions should be formulated in conjunction with the conclusion, making targeted approaches or suggestions rather than general approaches or suggestions. This paper only puts forward the measures of each urban agglomeration, authors can add the approaches of different cities in each urban agglomeration.

11. The discussion is not deep enough. No targeted policy recommendations are put forward for the research conclusions of this paper. It is suggested to refer to the following:

[2] Local and tele-coupling development between carbon emission and ecologic environment quality. Journal of Cleaner Production.

12. Some references, words and grammar have some problems in the format of the article. For example：

(1) in the keywords: barrier facto?

(2) Formula (2) is incorrectly in the format of article.

(3) Line 320, “led to increased government support” revises “led to increase ……”.

(4) Line 348, “3.3.Barrier factor analysis” differs from other subheadings title above.

(5) Line 367, “The following approach can be adopted”, you can write “The following approaches can be adopted”

(6) In the references. The 20th reference check please.

(7) Fig.2. need to be adjusted. Y-axis name and Y-axis scale need to be adjusted. The English font needs to be adjusted in the Fig.

and so on. The format of article can refer to the journal requirements.

Reviewer 2 Comments:

The authors analyze the vulnerability and obstacle factors of seven urban agglomerations along the Yellow River from the perspective of production-living-ecological space. Although the topic is practically meaningful, the manuscript lacks innovation and academic contribution. Firstly, examining urban vulnerability from the perspective of production-living-ecological space is not an original research idea, as numerous existing studies have adopted this approach. Secondly, the research methods are quite singular and superficial, relying mainly on basic techniques like entropy method and weighted summation to determine indicator weights and vulnerability indices, without employing more sophisticated and innovative assessment methods. Thirdly, the empirical analysis is overly simple and descriptive, without delving deeper into the inherent mechanisms driving the vulnerability of each urban agglomeration. Finally, the manuscript lacks targeted policy recommendations that provide clear guidance on how to reduce the vulnerability of urban agglomerations. In summary, the innovation and academic contribution of this study are inadequate. I would encourage the authors to further enhance the originality and depth of the research to increase its academic value. Extensive revisions and supplements are needed for the paper to meet the requirements for publication. Specific recommendations are as follows:

1 Introduction

(1)Line 92: The introduction lacks a comprehensive review of the existing literature in the field. It is essential to provide a clear articulation of how this study differentiates itself from prior research. The introduction should conclude with a paragraph that emphasizes the novel contributions and innovations of this research, thereby underscoring its significance in the academic discourse.

2 Materials and Methods

(1)Line 174: The choice of the proposed vulnerability measurement and barrier factor diagnosis models seems arbitrary. A more robust theoretical foundation is required to justify the selection of these models over potential alternatives. This would enhance the rationale and ensure that the methodology is grounded in established economic theories.

(2)Line 183: The introduction of the entropy method is cursory. A detailed exposition of its core concepts, strengths, and potential limitations is necessary. This will not only justify its selection for evaluating vulnerability indicators but also provide readers with a better understanding of its applicability and relevance.

3 Results

(1)Line 355: The barrier factor analysis, while identifying key dimensions, falls short in explaining the underlying mechanisms that influence vulnerability. A more in-depth discussion, rooted in economic theory, is imperative to shed light on the significance and interplay of these dimensions. This will also help in contextualizing the results within the broader economic framework.

4 Discussion

(1)Line 427: The discussion on policy implications is rather generic. Given the diverse barrier factors faced by different urban agglomerations, it is crucial to propose more nuanced and actionable recommendations tailored to the specific challenges of each agglomeration. This would make the findings more relevant for policymakers and practitioners.

(2)Line 465: The conclusion should acknowledge limitations and gaps of the current study, and discuss future improvements.

Reviewer 3 Comments:

1. The reasons for the selected indicators in the indicator system should be explained;

2. The discussion section should clearly indicate the marginal contribution of this article and compare it with existing research results;

3. Language expression needs improvement.

Reviewers' comments:

Reviewer's Responses to Questions

**Comments to the Author**

1. Is the manuscript technically sound, and do the data support the conclusions?

Reviewer #1: Yes

Reviewer #2: Yes

Reviewer #3: No

2. Has the statistical analysis been performed appropriately and rigorously? 

Reviewer #1: N/A

Reviewer #2: Yes

Reviewer #3: No

3. Have the authors made all data underlying the findings in their manuscript fully available?

Reviewer #1: Yes

Reviewer #2: No

Reviewer #3: No

4. Is the manuscript presented in an intelligible fashion and written in standard English?

Reviewer #1: No

Reviewer #2: No

Reviewer #3: No

5. Review Comments to the Author

Reviewer #1: Specific suggestions are as follows：

1. The structure of the abstract is incomplete. The author adds the suggestions or measures proposed and relevant data to revise the abstract.

2. Line 56 to 58, this paper says “However, compared to other regions, the ecological environment and economic……. urban agglomerations along the Yellow River”, Which articles did the author read to reach this conclusion, need to be explained, or use relevant data to illustrate this view.

3. Line 68 to 73, this paper summarizes the previous research, which is conducted from the quantitative scale, and evaluates it from the quality scale. I have read the full text, from your use of methods and selected indicators, it does not seem to reflect from perspective of quality. In the part of introduction, you can add content of research model.

4. In the part of literature review, the innovation of this paper and the contribution of previous studies have not been clearly expressed. It is suggested to refer to the following:

[1] The local coupling and telecoupling of urbanization and ecological environment quality based on multisource remote sensing data. Journal of Environmental Management.

5. In the part of data sources, be specific about where your data comes from, the data of each indicator comes from relevant documents to be clearly explained.

6. In the Methodology, whether there are evidences for the selection of all indicators, which need to explain. Why is the entropy method used to calculate the weight in this article? Are other methods of calculating index weight not feasible?

7. Line 164 to 218, why are there two identical subheadings?

8. Line 232 to 236, the explanation of three formulas is not clear, the reader may not understand it. You can write “where is…..”.

9. Line 310 to 347, the paper analyzes the rise of vulnerability from 2006 to 2010, 2011 to 2015 and 2016 to 2020. Therefore, the vulnerability of some urban agglomerations has decreased during these periods, should it also be specified?

10. Line 368 to 420, The proposed approaches or suggestions should be formulated in conjunction with the conclusion, making targeted approaches or suggestions rather than general approaches or suggestions. This paper only puts forward the measures of each urban agglomeration, authors can add the approaches of different cities in each urban agglomeration.

11. The discussion is not deep enough. No targeted policy recommendations are put forward for the research conclusions of this paper. It is suggested to refer to the following:

[2] Local and tele-coupling development between carbon emission and ecologic environment quality. Journal of Cleaner Production.

12. Some references, words and grammar have some problems in the format of the article. For example：

(1) in the keywords: barrier facto?

(2) Formula (2) is incorrectly in the format of article.

(3) Line 320, “led to increased government support” revises “led to increase ……”.

(4) Line 348, “3.3.Barrier factor analysis” differs from other subheadings title above.

(5) Line 367, “The following approach can be adopted”, you can write “The following approaches can be adopted”

(6) In the references. The 20th reference check please.

(7) Fig.2. need to be adjusted. Y-axis name and Y-axis scale need to be adjusted. The English font needs to be adjusted in the Fig.

and so on. The format of article can refer to the journal requirements.

Reviewer #2: 1. The reasons for the selected indicators in the indicator system should be explained;

2. The discussion section should clearly indicate the marginal contribution of this article and compare it with existing research results;

3. Language expression needs improvement.

Reviewer #3: The authors analyze the vulnerability and obstacle factors of seven urban agglomerations along the Yellow River from the perspective of production-living-ecological space. Although the topic is practically meaningful, the manuscript lacks innovation and academic contribution. Firstly, examining urban vulnerability from the perspective of production-living-ecological space is not an original research idea, as numerous existing studies have adopted this approach. Secondly, the research methods are quite singular and superficial, relying mainly on basic techniques like entropy method and weighted summation to determine indicator weights and vulnerability indices, without employing more sophisticated and innovative assessment methods. Thirdly, the empirical analysis is overly simple and descriptive, without delving deeper into the inherent mechanisms driving the vulnerability of each urban agglomeration. Finally, the manuscript lacks targeted policy recommendations that provide clear guidance on how to reduce the vulnerability of urban agglomerations. In summary, the innovation and academic contribution of this study are inadequate. I would encourage the authors to further enhance the originality and depth of the research to increase its academic value. Extensive revisions and supplements are needed for the paper to meet the requirements for publication. Specific recommendations are as follows:

1 Introduction

(1)Line 92: The introduction lacks a comprehensive review of the existing literature in the field. It is essential to provide a clear articulation of how this study differentiates itself from prior research. The introduction should conclude with a paragraph that emphasizes the novel contributions and innovations of this research, thereby underscoring its significance in the academic discourse.

2 Materials and Methods

(1)Line 174: The choice of the proposed vulnerability measurement and barrier factor diagnosis models seems arbitrary. A more robust theoretical foundation is required to justify the selection of these models over potential alternatives. This would enhance the rationale and ensure that the methodology is grounded in established economic theories.

(2)Line 183: The introduction of the entropy method is cursory. A detailed exposition of its core concepts, strengths, and potential limitations is necessary. This will not only justify its selection for evaluating vulnerability indicators but also provide readers with a better understanding of its applicability and relevance.

3 Results

(1)Line 355: The barrier factor analysis, while identifying key dimensions, falls short in explaining the underlying mechanisms that influence vulnerability. A more in-depth discussion, rooted in economic theory, is imperative to shed light on the significance and interplay of these dimensions. This will also help in contextualizing the results within the broader economic framework.

4 Discussion

(1)Line 427: The discussion on policy implications is rather generic. Given the diverse barrier factors faced by different urban agglomerations, it is crucial to propose more nuanced and actionable recommendations tailored to the specific challenges of each agglomeration. This would make the findings more relevant for policymakers and practitioners.

(2)Line 465: The conclusion should acknowledge limitations and gaps of the current study, and discuss future improvements.

6. PLOS authors have the option to publish the peer review history of their article (what does this mean?). If published, this will include your full peer review and any attached files.

Reviewer #1: No

Reviewer #2: No

Reviewer #3: No

---

## [Author Response · Author response to Decision Letter 0]

27 Dec 2023

Response to Reviewer 1 Comments on “The vulnerability assessment and obstacle factor analysis of urban agglomeration along the Yellow River in China from the perspective of production-living-ecological space” 

First of all, we would like to thank you for your valuable comments. All the suggestions are of great importance for us to improve the manuscript quality. Therefore, we have revised our manuscript according to the suggestions and made point-to-point responses to the reviewer’s comments, and correcting the writing language and grammar in the new manuscript.

Point 1: The structure of the abstract is incomplete. The author adds the suggestions or measures proposed and relevant data to revise the abstract.

Response 1: Thank you very much and agree with your question. We have refined the conclusions and added targeted policy recommendations in the abstract, which is already fully structured.

Here are the revised sentences.

Line28-43: The results reveal that the spatial differentiation characteristics of urban agglomeration vulnerability are significant. A clear three-level gradient distribution of high, medium, and low degrees is seen in the overall vulnerability; these correspond to the lower, middle, and upper reaches of the Yellow River Basin, respectively. The percentage of cities with higher and moderate levels of vulnerability did not vary from 2001 to 2020, while the percentage of cities with high levels of vulnerability did. The four dimensions of economic development, leisure and tourism, resource availability, and ecological pressure are the primary determinants of the urban agglomeration's vulnerability along the Yellow River. And the vulnerability factors of various urban agglomerations showed a significant evolutionary trend; the obstacle degree values have declined, and the importance of tourism and leisure functions has gradually increased. Based on the above conclusions, we propose several suggestions to enhance the quality of urban development along the Yellow River urban agglomeration. Including formulating a three-level development strategy, paying attention to ecological and environmental protection, developing domestic and foreign trade, and properly planning and managing the tourism industry.

Point 2: Line 56 to 58, this paper says “However, compared to other regions, the ecological environment and economic……. urban agglomerations along the Yellow River”, Which articles did the author read to reach this conclusion, need to be explained, or use relevant data to illustrate this view.

Response 2: Thank you for your suggestions. Two high-quality articles have been cited to support the scientific validity of this conclusion. We have added these articles to the references.

Point 3: Line 68 to 73, this paper summarizes the previous research, which is conducted from the quantitative scale, and evaluates it from the quality scale. I have read the full text, from your use of methods and selected indicators, it does not seem to reflect from perspective of quality. In the part of introduction, you can add content of research model.

Response 3：Thank you for your suggestions. We have added a specific explanation of the word “quality” in the Introduction. The "quality" mentioned in this article only refers to the quality of production, the quality of living and ecological space. We assess the quality of urban development primarily by measuring the vulnerability of urban agglomerations along the Yellow River. Based on the perspectives of production, life and ecological space, we established a comprehensive assessment index system, including 9 first-level indicators and 27 second-level indicators, to measure the vulnerability of urban agglomerations along the Yellow River and thereby evaluate its urban development quality. Subsequently, a barrier factor diagnostic model was used to determine the main factors affecting the development quality of urban agglomerations, as well as the evolutionary trends of the influencing factors. We briefly summarize the model in the introduction, and the specific indicator system and obstacle model construction are explained in detail in the methods chapter.

Point 4: In the part of literature review, the innovation of this paper and the contribution of previous studies have not been clearly expressed. It is suggested to refer to the following:

[1] The local coupling and telecoupling of urbanization and ecological environment quality based on multisource remote sensing data. Journal of Environmental Management.

Response 4：Thank you very much and agree with your question. We have carefully read the article you recommended, and it is of great significance for reference. Subsequently, we refer to it in the introduction part to explain the innovations and contributions of this paper in detail, and add the structural description of the full text, as follows:

Line111-134: The innovations and contributions of this article are: (1) Taking the seven urban agglomerations along the Yellow River as the research object, a better understanding of the integrity and systematicity of the Yellow River urban agglomerations will help promote integration and coordination among cities. The research results have certain reference significance for urban management planning and sustainable development in other similar regions around the world. (2) On the basis of existing research, we establish an index system for evaluating vulnerability that is more comprehensive and efficiently evaluates the vulnerability of the urban agglomerations along the Yellow River between 2001 and 2020. From the perspective of production-living-ecological space, the original assessment index system is synthesized, and new indicators are creatively introduced in conjunction with the real conditions of the research region. (3) We analyze the vulnerability assessment results from the perspective of the overall vulnerability and spatial variability of the urban agglomeration. Furthermore, the obstacle factor model was used to identify the influencing factors, and then the evolution trend of the obstacle factors was analyzed. Based on these, targeted management measures are proposed for the urban agglomeration along the Yellow River.

The following is the arrangement of the paper's other sections: Section 2 is the materials, which describe the study area and data sources. Section 3 presents the methodology, including the vulnerability assessment model, vulnerability measurement index model, and obstacle factor diagnostic model. Section 4 is the results and also includes overall vulnerability analysis, variability analysis within urban agglomerations, and barrier factor analysis. Section 5 discusses the results. Section 6 provides the conclusions and outlook of the paper.

Point 5: In the part of data sources, be specific about where your data comes from, the data of each indicator comes from relevant documents to be clearly explained.

Response 5：Thank you for your suggestions. In the manuscript, we have presented the specific sources and documentation of indicator data at each level.

Here are the revised sentences.

Line187-201: This article focuses on assessing the vulnerability of seven urban agglomerations in the Yellow River Basin from 2001 to 2020. To conduct this assessment, the researchers selected 9 first-level indicators and 27 second-level indicators. These indicators were used to analyze the vulnerability factors and their evolution trends through a barrier factor diagnostic model. The data for the indicators related to agricultural production, social security services, ecological pressure, and ecological governance were obtained from “the Statistical Yearbooks” of each province in the Yellow River Basin. The data for indicators related to non-agricultural production, economic development, living services, and resource supply were sourced from “the China Urban Statistical Yearbook”. The data at the tourism and leisure functions level came from “the Statistical Bulletins” of various cities. For the indicators related to per capita water resources at the resource supply level, the data came from “the Water Resources Bulletin”. The carbon emission data per 10,000yuan of GDP were obtained from Chen’s research [28]. In cases where some data were missing, interpolation methods were used to supplement the missing values.

Point 6: In the Methodology, whether there are evidences for the selection of all indicators, which need to explain. Why is the entropy method used to calculate the weight in this article? Are other methods of calculating index weight not feasible?

Response 6: Thank you very much and agree with your question. In the index system construction, we have explained in detail the selection of indicators for production, life, and ecological space quality, and innovatively added new indicators. Subsequently, in the entropy method section, we explained the reason why this article chose the entropy method for indicator weighting. 

Here are the revised sentences.

Line205-240: From the perspective of production-living-ecological space, Thie article builds a comprehensive evaluation index system based on the concepts of representativeness, comprehensiveness, scientificity, rationality, and operability, considering the findings of previous research. Every detail of the Yellow River urban agglomeration's actual situation is taken into account. Three views are used in the construction of the comprehensive assessment index system: the production space quality, the living space quality, and the ecological space quality. In terms of the selection of production space quality and living space quality indicators, Zhang et al. 's research on production-life-ecological function of urban agglomeration in the middle reaches of the Yangtze River is mainly referred to [29]. For the selection of indicators for ecological space quality, Fang et al.'s research on the spatial differentiation of urban fragility in China is primarily consulted [30]. Although existing research has established a vulnerability evaluation index system, there are still some shortcomings that need to be addressed. For instance, existing indicators lack comprehensiveness when measuring agricultural production, as they only evaluate the overall output of agriculture without considering agricultural machinery power. Similarly, when measuring non-agricultural production, they only take industrial production into account, while ignoring the tertiary industry. Moreover, indicators for measuring the quality of living services only consider economic income and basic security, without taking into account urban population pressure, a crucial factor affecting citizens’ lives. Lastly, the indicators for measuring the quality of living space do not keep up with the times, as they do not include the important impact of tourism and leisure on residents’ quality of life. 

Therefore, this study proposes four improvements. Firstly, to more accurately evaluate agricultural production capacity, new indicators are added, such as Per unit of land gross power of agricultural machinery. Secondly, to fully assess non-agricultural production capability, per unit of land output of secondary and tertiary industries are included. Thirdly, a new dimension related to tourism and leisure is added in the measurement of living space quality. This is measured using indicators such as per unit of land domestic tourist arrivals and per unit of land domestic tourism revenue, providing a more accurate representation of residents’ quality of life in the Yellow River urban agglomeration. Finally, minor adjustments are made to the wording of indicators for ecological space quality based on the specific characteristics of the Yellow River urban agglomeration. 

In conclusion, a vulnerability assessment index system for the Yellow River urban agglomeration was constructed, including 9 first-level indicators and 27 second-level indicators (as shown in Table 2).

Line245-256: The entropy method is a widely used weight allocation method in multi-criteria decision making, aimed at determining the relative importance of each index or criterion in the decision-making process. This method evaluates the importance of each factor by assessing the entropy of each index based on the concept of information entropy and assigns the corresponding weight. Compared to methods such as the Delphi method, expert survey method, and analytic hierarchy process, the entropy method excludes subjective factors in the weighting process, resulting in more objective index weights [31]. This eliminates the influence of varying data, allowing for the comparison of the evaluation of objects using a unified standard over the years. To calculate and evaluate the vulnerability of urban agglomerations along the Yellow River, this paper first utilizes the range method to standardize the data for each index, then applies the entropy method. The specific calculation process follows:

Point 7: Line 164 to 218, why are there two identical subheadings?

Response 7: Thank you for your suggestions. In the manuscript, we have replaced the second subheading “Vulnerability assessment model” with “Obstacle factor diagnostic model”. 

Point 8: Line 232 to 236, the explanation of three formulas is not clear, the reader may not understand it. You can write “where is…..”.

Response 8: Thank you very much and agree with your question. We have explained these three formulas in more detail to help readers understand them better, specifically:

Line305-319:

(1) The factor contribution degree:

 (7)

Where Fij is the factor contribution degree, wi is the weight of the ith subsystem, and pij is the weight of the jth indicator of the ith subsystem.

(2) The index deviation degree:

 (8)

Where Vi is index deviation degree, and Xij is the standardized value of a single indicator.

(3) The obstacle degree:

 (9)

 Where Cij is obstacle degree, and the larger the value, the higher the degree of obstacle that the indicator has for the vulnerability of the urban agglomeration along the Yellow River.

Point 9: Line 310 to 347, the paper analyzes the rise of vulnerability from 2006 to 2010, 2011 to 2015 and 2016 to 2020. Therefore, the vulnerability of some urban agglomerations has decreased during these periods, should it also be specified?

Response 9: Thank you very much and agree with your question. We have provided a more specific explanation of the decline in vulnerability in Yinchuan from 2006 to 2010, Xinxiang from 2011 to 2015, and Lvliang from 2016 to 2020.

Here are the revised sentences.

Line429-439: The main reason is that the construction of the urban economic belt along the Yellow River was proposed in 2005. This policy was carried out with Yinchuan as the center, which brought a lot of policy preferences and resource support to the development of Yinchuan. Yinchuan has invested a lot of investment in urban construction and built an urban transportation network in accordance with the concept of "near connections, distant connections, and external expansion". At the same time, Yinchuan actively implements the strategy of moving the city center westward, accelerating the development of new areas and the renovation of old cities. In addition, a large number of landscaping and wetland protection projects have been implemented to promote the improvement of the urban ecological environment and reduce the vulnerability of Yinchuan.

Line454-467: This shows that Ordos has not taken into account the ecological environment while rapidly developing its economy. It is an extensive development at the expense of the environment, which is not conducive to the sustainable development of the city. But on the other hand, there were also cities whose vulnerability declined during this period, namely Xinxiang in the Central Plains urban agglomeration, which dropped from high vulnerability to moderate vulnerability. At the first annual meeting of the Low Carbon China Forum in 2010, Xinxiang was awarded the title of “Actively Developing Low-Carbon Economic City”, the only city in the Central Plains urban agglomeration to be awarded the title. Following that, Xinxiang made a concerted effort to create a low-carbon circular economy and placed a high value on technological advancement and industrial innovation. Simultaneously, it employs multifaceted spatial planning techniques to enhance the low-carbon spatial arrangement, mitigate urban susceptibility, and promote the city's sustainable development.

Line473-484: The period from 2016 to 2020 is China's "Thirteenth Five-Year Plan" period. It is a stage when China's economic development changes from high-speed development to high-quality development. Urban development shifted from solely focusing on economic growth in terms of speed and scale to emphasizing coordinated development across the economy, society, and environment. During the study period, these cities vigorously improved resource utilization efficiency, built environmentally friendly industries, and focused on the maintenance and restoration of the ecological environment, which improved the quality of urban tertiary space and reduced urban vulnerability. Moreover, due to the impact of the COVID-19 epidemic in 2020, human activities have decreased, and the pressure on infrastructure and the environment has decreased. This results in ecological restoration and reduced urban vulnerability.

Point 10: Line 368 to 420, The proposed approaches or suggestions should be formulated in conjunction with the conclusion, making targeted approaches or suggestions rather than general approaches or suggestions. This paper only puts forward the measures of each urban agglomeration, authors can add the approaches of different cities in each urban agglomeration.

Response 10: Thank you very much and agree with your question. Based on the research conclusions of this article, in order to improve the quality of urban development along the Yellow River urban agglomeration, we put forward the following targeted suggestions: 

Line735-788:

(1) Formulate a three-level development strategy. Based on the vulnerability of each urban agglomeration, local governments should formulate urban agglomeration development plans based on local conditions. Highly vulnerable urban agglomerations should implement a green circular economy model, moderately vulnerable cities need to focus on optimizing industrial structure and upgrading production technology, and low-vulnerability urban agglomerations should accelerate opening up and industrial regional cooperation. In particular, the urban agglomeration of Shandong Peninsula ought to initiate ecological restoration initiatives and cultivate a varied energy framework. The creation of a circular economy model and bolstering the growth of international commerce scale should be the priorities of the Central Plains urban agglomeration. The metropolitan agglomerations of Yu, Baotou, and Hubei should increase their investments in research and technology, introduce more professional talents, and support the modernization of production methods and the optimization of the industrial structure. The urban agglomerations of Jinzhong, Guanzhong Plain, Lanxi, and Ningxia along the Yellow River are expected to foster coordinated development and strengthen regional collaboration.

(2) Pay attention to ecological and environmental protection. Local governments should coordinate and reconcile the contradiction between ecology and development and adhere to sustainable development. For central cities with high vulnerability, some functions of the city should be distributed to surrounding cities to alleviate the pressure on the urban ecological environment; while surrounding cities should strengthen urban greening construction and promote regional cooperation to form an urban ecosystem and realize inter-city collaborative development. At the same time, local governments should establish ecological restoration projects. For upstream urban agglomerations such as Ningxia urban agglomeration, the construction of green ecological corridors should be accelerated to ensure the integrity and stability of the ecosystem in the Yellow River Basin. For urban agglomerations along the lower reaches of the Yellow River such as the Central Plains urban agglomeration, it is necessary to Continue to promote pollution prevention and control, carbon reduction and emission reduction, and enhance the ecological barrier function of the Yellow River Basin.

(3) Develop domestic and foreign trade. Due to locational constraints, the level of opening-up of urban agglomerations along the Yellow River is relatively effective. In this regard, governments at all levels should increase infrastructure investment and construction, establish a high-quality transportation network throughout the Yellow River Basin, and promote urban interconnection. At the same time, we should take Shandong Peninsula as the window of regional foreign trade, and vigorously develop export-oriented industries and carry out import and export business. And provide preferential policies to attract regional investment, improve the level of foreign trade development. 

(4) Properly plan and manage the tourism industry. Regional governments should give full play to the leading role of the tourism industry. Based on the regional Yellow River cultural advantages, relevant policies have been introduced to develop tourism into a regional advantageous industry. Urban agglomerations with high tourism and leisure vulnerability should optimize tourism resource management, enhance tourism resource utilization efficiency, and develop low-carbon tourism products and projects to reduce energy consumption in the tourism industry. For urban agglomerations with a low level of tourism industry development, they should promote technological upgrading and structural optimization and upgrading of the tourism industry by accelerating the introduction of tourism professionals and promoting talent training and information exchange. At the same time, each urban agglomeration should also vigorously promote urban tourism cooperation. This is the main way to promote the complementary advantages of the tourism industry in the upper, middle and lower reaches of the Yellow River Basin and narrow regional differences. The ultimate goal is to promote coordinated regional development.

Point 11: The discussion is not deep enough. No targeted policy recommendations are put forward for the research conclusions of this paper. It is suggested to refer to the following:

[2] Local and tele-coupling development between carbon emission and ecologic environment quality. Journal of Cleaner Production.

Response 11: Thank you for your suggestions. We have read the article you recommended to us in detail and it is very logical and academic. The research conclusions of our article have been deeply optimized with reference to this article, and mainly include the following four points:

(1) The spatial differentiation characteristics of urban agglomeration vulnerability are significant. A clear three-level gradient distribution of high, medium, and low degrees is seen in the overall vulnerability; these correspond to the lower, middle, and upper reaches of the Yellow River Basin, respectively. The specific vulnerability levels are: Shandong Peninsula urban agglomeration > Central Plains urban agglomeration > Hubao and Eyu urban ag-glomeration > Jinzhong urban agglomeration > Guanzhong Plain urban agglomeration > Lanxi urban agglomeration > Ningxia urban agglomeration. 

(2) The percentage of cities with higher and moderate levels of vulnerability did not vary from 2001 to 2020, while the percentage of cities with high levels of vulnerability did. The central cities of each urban agglomeration are always highly vulnerable, and the surrounding cities are slightly less vulnerable than them. 

(3) There are differences in the vulnerability barrier factors of each urban agglomeration along the Yellow River. The four dimensions of economic development, leisure and tourism, resource availability, and ecological pressure are the primary determinants of the urban agglomeration's vulnerability along the Yellow River.

(4) From 2001 to 2020, the vulnerability factors of various urban agglomerations showed a significant evolutionary trend, the obstacle degree values have declined, and the importance of tourism and leisure functions has gradually increased.

Point 12: Some references, words and grammar have some problems in the format of the article. For exampleï¼š

(1) in the keywords: barrier facto?

(2) Formula (2) is incorrectly in the format of article.

(3) Line 320, “led to increased government support” revises “led to increase ……”.

(4) Line 348, “3.3.Barrier factor analysis” differs from other subheadings title above.

(5) Line 367, “The following approach can be adopted”, you can write “The following approaches can be adopted”

(6) In the references. The 20th reference check please.

(7) Fig.2. need to be adjusted. Y-axis name and Y-axis scale need to be adjusted. The English font needs to be adjusted in the Fig.

and so on. The format of article can refer to the journal requirements.

Response 12: Thank you very much and agree with your question. We have improved the references, vocabulary and grammar of this article as follow:

(1) We have replaced the keyword “barrier facto” with “barrier factor”.

(2) We have revised formula (2) as follows:

(3) Line431-432, We have replaced “led to increased government support” with “This policy was carried out with Yinchuan as the center, which brought a lot of policy preferences and resource support to the development of Yinchuan”.

(4) Line485, We have replaced “3.3. Barrier factor analysis” with “Obstacle factor analysis”, and replaced “Analysis of variability within urban agglomeration” with “Difference analysis within urban agglomerations”

(5) Line507, We have replaced “The following approach can be adopted” with “The following approaches can be adopted”.

(6) Line865, We have checked the 20th reference and corrected it.

(7) We have replaced the name of the Y-axis “Vulnerability” with “Vulnerability index”, and adjusted the scale of the Y-axis.

Response to Reviewer 2 Comments on “The vulnerability assessment and obstacle factor analysis of urban agglomeration along the Yellow River in China from the perspective of production-living-ecological space” 

First of all, thank you for your valuable comments. These suggestions are of great significance for us to improve the quality of manuscripts. Indeed, some scholars in the field have examined urban vulnerability from the perspective of three-dimensional space, which also proves the reliability of the model. However, no scholar has yet applied this model to the study of the urban agglomeration along the Yellow River. We were inspired by other people's research and optimized and innovated the existing vulnerability evaluation index system based on previous research. Subsequently, the obstacle factor diagnostic model is used to identify the obstacle factors that affect the vulnerability of the urban agglomeration along the Yellow River and analyse their evolution trends, thereby providing targeted policy recommendations for its development. The results are more robust and scientific. Secondly, although the research method of this article is relatively simple, considering the actual development situation of the urban agglomeration along the Yellow River, the method used in this article is sufficient to identify its main obstacles and analyse evolutionary trends. This has been able to provide quick and effective guidance for its development, and we will conduct more in-depth research on this basis in the future. Finally, thank you again for your valuable suggestions. We have made a lot of detailed revisions and additions to the reviewers' suggestions, and increased the innovation and academic contribution of this article. We responded to the reviewers’ comments point-to-point and made revisions to the writing language and grammar in the new manuscript

Point 1: Introduction

(1)Line 92: The introduction lacks a comprehensive review of the existing literature in the field. It is essential to provide a clear articulation of how this study differentiates itself from prior research. The introduction should conclude with a paragraph that emphasizes the novel contributions and innovations of this research, thereby underscoring its significance in the academic discourse.

Response 1: Thank you very much and agree with your question. After reading a large number of articles in related fields, we made a clear statement of the existing literature and highlighted the innovations and contributions of this article at the end of the introduction. At the same time, we also explain the paragraph structure of the article.

Here are the revised sentences.

Line73-85: Urban vulnerability research has become a crucial scientific method for exploring the harmonious symbiosis of human and environmental systems [13]. However, most of the current studies are based on the quantity and scale of cities and lack a comprehensive assessment from the perspective of the quality of urban development. Therefore, we aim to develop a comprehensive evaluation system for urban vulnerability to serve as a reference for assessing the overall quality of urban development and promoting sustainable development for both people and the environment. In recent years, with the gradual popularization of the "three pillars" concept in the international community, academics have increasingly focused on the study of territorial space from a production-living-ecological perspective [23,24]. The integrated spatial layout of production-living-ecological space plays a crucial role in harmonizing the relationship between people and the land and provides a methodology for improving the quality of urban development and reducing urban vulnerability.

Line111-134: The innovations and contributions of this article are: (1) Taking the seven urban agglomerations along the Yellow River as the research object, a better understanding of the integrity and systematicity of the Yellow River urban agglomerations will help promote integration and coordination among cities. The research results have certain reference significance for urban management planning and sustainable development in other similar regions around the world. (2) On the basis of existing research, we establish an index system for evaluating vulnerability that is more comprehensive and efficiently evaluates the vulnerability of the urban agglomerations along the Yellow River between 2001 and 2020. From the perspective of production-living-ecological space, the original assessment index system is synthesized, and new indicators are creatively introduced in conjunction with the real conditions of the research region. (3) We analyze the vulnerability assessment results from the perspective of the overall vulnerability and spatial variability of the urban agglomeration. Furthermore, the obstacle factor model was used to identify the influencing factors, and then the evolution trend of the obstacle factors was analyzed. Based on these, targeted management measures are proposed for the urban agglomeration along the Yellow River.

The following is the arrangement of the paper's other sections: Section 2 is the materials, which describe the study area and data sources. Section 3 presents the methodology, including the vulnerability assessment model, vulnerability measurement index model, and obstacle factor diagnostic model. Section 4 is the results and also includes overall vulnerability analysis, variability analysis within urban agglomerations, and barrier factor analysis. Section 5 discusses the results. Section 6 provides the conclusions and outlook of the paper.

Point 2: Materials and Methods

(1) Line 174: The choice of the proposed vulnerability measurement and barrier factor diagnosis models seems arbitrary. A more robust theoretical foundation is required to justify the selection of these models over potential alternatives. This would enhance the rationale and ensure that the methodology is grounded in established economic theories.

(2) Line 183: The introduction of the entropy method is cursory. A detailed exposition of its core concepts, strengths, and potential limitations is necessary. This will not only justify its selection for evaluating vulnerability indicators but also provide readers with a better understanding of its applicability and relevance.

Response 2: 

(1) Thank you for your suggestions. The choice of the proposed vulnerability measurement and barrier factor diagnosis models is not arbitrary, and we chose it after referring to many articles. We have explained the reasons and theoretical basis for using this model in the article. In addition, a lot of optimizations has been done to the methods part of the article to ensure its scientific validity.

(2) Thank you very much and agree with your question. In the manuscript, we add a description of the entropy method to justify choosing it to evaluate vulnerability indicators. The specific contents are as follows:

Line245-256: The entropy method is a widely used weight allocation method in multi-criteria decision making, aimed at determining the relative importance of each index or criterion in the decision-making process. This method evaluates the importance of each factor by assessing the entropy of each index based on the concept of information entropy and assigns the corresponding weight. Compared to methods such as the Delphi method, expert survey method, and analytic hierarchy process, the entropy method excludes subjective factors in the weighting process, resulting in more objective index weights [31]. This eliminates the influence of varying data, allowing for the comparison of the evaluation of objects using a unified standard over the years. To calculate and evaluate the vulnerability of urban agglomerations along the Yellow River, this paper first utilizes the range method to standardize the data for each index, then applies the entropy method. The specific calculation process follows:

Point 3: Results

(1) Line 355: The barrier factor analysis, while identifying key dimensions, falls short in explaining the underlying mechanisms that influence vulnerability. A more in-depth discussion, rooted in economic theory, is imperative to shed light on the significance and interplay of these dimensions. This will also help in contextualizing the results within the broader economic framework.

Response 3: Thank you very much and agree with your question. We must admit that the analysis of barrier factors previously in the manuscript was indeed not in depth enough. We divide the barrier factor analysis part into key barrier factor analysis and barrier factor evolution analysis to explain the impact mechanism of vulnerability more scientifically. In addition, a more in-depth discussion based on economic theory is added to conduct a multi-dimensional analysis of the vulnerability of the urban agglomeration along the Yellow River and its influencing factors. 

Here are the added sentences.

Line569-704:

Obstacle factor evolution analysis

In order to gain a deeper understanding of the evolution trend of the vulnerability of urban agglomerations along the Yellow River, we next conducted the following comparative analysis of the obstacle factors of each urban agglomeration in 2001 and 2002.

Firstly, for the urban agglomeration of the first echelon of vulnerability. During the study period, the ranking of energy consumption per 10,000yuan of GDP of the Shandong Peninsula urban agglomeration dropped from third to sixth, and the obstacle degree dropped from 5.92 to 5.21. Similarly, the factor ranking of industrial wastewater discharge of 10,000yuan of GDP decreased from the sixth to the tenth, and its obstacle degree decreased from 5.51 to 4.82. It can be seen that the Shandong Peninsula urban agglomeration has achieved certain results in energy structure adjustment and environmental governance. However, there is insufficient supply of clean energy, and there is a high dependence on high-carbon energy sources, leading to high carbon emissions per 10,000yuan of GDP. Regarding the Central Plains urban agglomeration, the disappearance of the obstacle factor per capita GDP indicates that during the study period, the overall economic development of the Central Plains urban agglomeration was strong, and productivity significantly improved. Furthermore, it should be mentioned that in 2020, both the Shandong Peninsula urban agglomeration and the Central Plains urban agglomeration saw an increase in the obstacle factors, i.e., per unit of land domestic tourist arrivals and per unit of land domestic tourism revenue, which are the key indicators that we focused on in this study. This observation implies that in the course of the development and evolution of urban agglomerations, the focus of development has shifted from primary and secondary industries towards the tertiary industry. The rapid development of the tertiary industry, represented by tourism, has put significant pressure on urban transportation, energy consumption, and the ecological environment. As a result, the vulnerable first-tier urban agglomerations should tighten up on planning and control in the tourism industry. The development of low-carbon tourist projects and goods, as well as the establishment of a strong tourism management system and efficient resource management, are imperative. By reducing energy consumption in the tourism industry, these urban agglomerations can promote sustainable urban development.

Secondly, for the urban agglomeration of the second echelon of vulnerability. During the study period, obstacle factors related to the tourism and leisure function dimensions were added to both the Hubao and Eyu urban agglomeration in 2020. These included per unit of land domestic tourist arrivals and per unit of land domestic tourism revenue, which came in at number four and sixth, respectively, with 5.19 and 4.94 obstacle degrees. Other obstacle factors did not change significantly. This shows that with the development of urban agglomerations, the significance of tourism and leisure functions is growing. From a specific perspective, the number of domestic tourists and domestic tourism income in Erdos and Yulin, within the Hubao Eyu City cluster, are relatively low. There is also limited tourism exchange between these cities. As a result, the overall level of tourism development in these cities is low, which makes it difficult to meet the tourism and leisure needs of local residents and optimize the urban industrial structure. Therefore, for the second level of vulnerable city clusters, it is essential to strengthen the optimization and upgrading of the industrial structure. This can be achieved by promoting the development of the tourism industry and facilitating tourism exchange and collaboration between cities. These efforts will contribute to the transformation and development of the city clusters, providing them with the necessary impetus for growth.

Thirdly, regarding the third echelon of vulnerable urban agglomeration s, including the Ningxia urban agglomeration, the Jinzhong urban agglomeration, the Guanzhong Plain urban agglomeration, and the Lanxi urban agglomeration, the ranking of obstacle factors has remained relatively stable compared to the previous two echelons. These urban agglomerations are generally more stable. From the perspective of obstacle degree, it can be divided into two types for analysis: The first type includes Ningxia urban agglomeration and Jinzhong urban agglomeration along the Yellow River. The vulnerability levels of production, living, and ecological spaces in these two city clusters have mostly decreased. This indicates that, with the Ningxia urban agglomeration as the main focus, efforts to promote coordinated development of the Yellow River Ecological Economic Belt and the Northern Green Development Zone have achieved certain results. The quality of urban construction and development has improved, and the regional spatial pattern continues to be optimized. On the contrary, the other types are Guanzhong Plain urban agglomeration and Lanxi urban agglomeration, and the obstacle degree of each factor in them is mostly increased. It can be seen that although these cities have lower vulnerability, they still face different challenges in the development process. Development planning needs to be tailored to local conditions, and regional industrial cooperation needs to be further improved.

Discussion

Based on the above research results, we compare this article with previous research results to illustrate the scientific nature and marginal contribution of this article. 

Overall, the urban agglomeration along the Yellow River has a remarkable three-level gradient differential in terms of vulnerability. This characteristic is the same as earlier research, and closely corresponds to the overall vulnerability gradient differentiation of Chinese urban agglomerations [13]. This demonstrates that, as a whole, the degree of vulnerability of the urban agglomeration along the Yellow River examined in this research is congruent with the real circumstances in China. Because the gradient of vulnerability at all levels is generally consistent with previous research on urban areas in the upper, middle, and lower ranges of the Yellow River, it suggests that the study is scientific. On the other hand, the total vulnerability of the urban agglomeration along the Yellow River indicates an increasing trend, based on the specific value of the obstacle degree. The percentage of cities with high susceptibility rises, while the percentage of cities with moderate or high vulnerability stays constant. This is consistent with previous research conclusions on urban agglomerations in the Yangtze River Basin [34,35], Pearl River Basin [36] and other regions, but is contrary to research conclusions in northeast China [37], central Yunnan [38], Zhangjiakou [39] and other regions of China. It is evident that the vulnerability of the urban agglomeration along the Yellow River has a particular influencing mechanism.

Urban agglomeration is a coupled system of economy, society and environment. Prior research on the Yellow River Basin has mostly concentrated on environmental and economic issues [40-43]. The findings indicate that the primary determinants of the urban agglomeration's growth along the Yellow River are resource availability, ecological environment, and economic development. The research conclusions of this article have a high degree of fit with it. However, existing studies lack the perspective of living space quality. Although there are studies on social and people's livelihood aspects, they are more focused on policy support and lack research on urban tourism and leisure functions [44,45]. It can be seen that the existing research on the quality of urban development in the Yellow River Basin is incomplete. Therefore, this paper constructs a multi-dimensional index system to comprehensively evaluate the evolutionary characteristics of vulnerability barrier factors in the urban agglomeration along the Yellow River. Compared with existing evaluation methods, this index system is more systematic and scientific.

Contrary to earlier research, the study's findings indicate that the influence of leisure and tourism activities on the vulnerability of urban agglomerations along the Yellow River has grown dramatically and is especially significant. We proceed from an economic perspective based on supply and demand effects. In terms of demand, with the continuous development of China's economy and the continuous increase in national income, the demand for tourism is becoming increasingly strong. This has led to a continuous increase in the number of tourists per region, which has put forward higher-level requirements for the development of tourism in urban agglomerations along the Yellow River; At the same time, as China's economy shifts from high-speed development to high-quality development, the country has attached great importance to the protection of the Yellow River culture and the development of the tourism industry since 2020. Higher requirements have been placed on the average tourism income of urban agglomerations along the Yellow River, which has led to the increasingly prominent importance of tourism and leisure functions in the development of urban vulnerability. In terms of supply, urban agglomerations along the Yellow River mainly focused on the development of primary and tertiary industries in the past, with the tertiary industry accounting for a relatively low proportion. The tourism supply was seriously insufficient and difficult to meet social needs. Although the industrial structure has improved in recent years and the tertiary industry has developed, the development level of the tourism industry is not high, mostly at the expense of consuming resources and damaging the environment. The average number of tourists per area and the average tourism income are not high, which is extremely detrimental to the high-quality development of the city and increases the vulnerability of the urban agglomeration along the Yellow River.

In summary, the marginal contributions of this article are as follows: This article improves the existing indicator system and conducts a scientific analysis of the overall vulnerability and obstacle factor evolution of the urban agglomeration along the Yellow River. The high-quality development level of urban agglomerations along the Yellow River is studied not only from economic and environmental aspects, but also from social aspects. It was found that the tertiary industry, specifically tourism and leisure functions, is particularly important to urban vulnerability. Theoretically, this article enriches previous research on the Yellow River Basin and provides a new perspective on tourism and leisure research. In practice, the research results can guide the planning and management of urban agglomerations along the Yellow River, and provide reference for their sustainable development.

Point 3: Discussion

(1) Line 427: The discussion on policy implications is rather generic. Given the diverse barrier factors faced by different urban agglomerations, it is crucial to propose more nuanced and actionable recommendations tailored to the specific challenges of each agglomeration. This would make the findings more relevant for policymakers and practitioners.

(2) Line 465: The conclusion should acknowledge limitations and gaps of the current study, and discuss future improvements.

Response 3: 

(1) Thank you for your suggestions. We have changed the discussion part to the conclusion. In the above analysis of the results, we have added detailed and specific policy impact statements to illustrate the mechanism of vulnerability of each urban agglomeration. In addition, in the conclusion section, we put forward more detailed and actionable suggestions for the vulnerability gradient and barrier factors of each urban agglomeration.

Here are the added policy suggestions.

Line735-788: (1) Formulate a three-level development strategy. Based on the vulnerability of each urban agglomeration, local governments should formulate urban agglomeration development plans based on local conditions. Highly vulnerable urban agglomerations should implement a green circular economy model, moderately vulnerable cities need to focus on optimizing industrial structure and upgrading production technology, and low-vulnerability urban agglomerations should accelerate opening up and industrial regional cooperation. In particular, the urban agglomeration of Shandong Peninsula ought to initiate ecological restoration initiatives and cultivate a varied energy framework. The creation of a circular economy model and bolstering the growth of international commerce scale should be the priorities of the Central Plains urban agglomeration. The metropolitan agglomerations of Yu, Baotou, and Hubei should increase their investments in research and technology, introduce more professional talents, and support the modernization of production methods and the optimization of the industrial structure. The urban agglomerations of Jinzhong, Guanzhong Plain, Lanxi, and Ningxia along the Yellow River are expected to foster coordinated development and strengthen regional collaboration.

(2) Pay attention to ecological and environmental protection. Local governments should coordinate and reconcile the contradiction between ecology and development and adhere to sustainable development. For central cities with high vulnerability, some functions of the city should be distributed to surrounding cities to alleviate the pressure on the urban ecological environment; while surrounding cities should strengthen urban greening construction and promote regional cooperation to form an urban ecosystem and realize inter-city collaborative development. At the same time, local governments should establish ecological restoration projects. For upstream urban agglomerations such as Ningxia urban agglomeration, the construction of green ecological corridors should be accelerated to ensure the integrity and stability of the ecosystem in the Yellow River Basin. For urban agglomerations along the lower reaches of the Yellow River such as the Central Plains urban agglomeration, it is necessary to Continue to promote pollution prevention and control, carbon reduction and emission reduction, and enhance the ecological barrier function of the Yellow River Basin.

(3) Develop domestic and foreign trade. Due to locational constraints, the level of opening-up of urban agglomerations along the Yellow River is relatively effective. In this regard, governments at all levels should increase infrastructure investment and construction, establish a high-quality transportation network throughout the Yellow River Basin, and promote urban interconnection. At the same time, we should take Shandong Peninsula as the window of regional foreign trade, and vigorously develop export-oriented industries and carry out import and export business. And provide preferential policies to attract regional investment, improve the level of foreign trade development. 

(4) Properly plan and manage the tourism industry. Regional governments should give full play to the leading role of the tourism industry. Based on the regional Yellow River cultural advantages, relevant policies have been introduced to develop tourism into a regional advantageous industry. Urban agglomerations with high tourism and leisure vulnerability should optimize tourism resource management, enhance tourism resource utilization efficiency, and develop low-carbon tourism products and projects to reduce energy consumption in the tourism industry. For urban agglomerations with a low level of tourism industry development, they should promote technological upgrading and structural optimization and upgrading of the tourism industry by accelerating the introduction of tourism professionals and promoting talent training and information exchange. At the same time, each urban agglomeration should also vigorously promote urban tourism cooperation. This is the main way to promote the complementary advantages of the tourism industry in the upper, middle and lower reaches of the Yellow River Basin and narrow regional differences. The ultimate goal is to promote coordinated regional development.

(2) Thank you very much and agree with your question. In the manuscript, we have added the limitations of the current research and the prospects for the future, as follows:

Line787-799：Of course, we must admit that this study has some shortcomings. Firstly, due to limitations in data availability and processing methods, this study only calculated a few factors reflecting tourism and leisure functions. More indicators can be added to future research for more comprehensive measurements. Secondly, although we have conducted a more in-depth analysis of the vulnerability and vulnerability factors of the urban agglomeration along the Yellow River, there is a lack of further discussion on the spatial impact relationship of each urban agglomeration. In future research, we can learn from more related spatial econometrics techniques to conduct more in-depth horizontal spatial research on the urban agglomeration along the Yellow River, thereby providing more reference suggestions for the coordinated development of the urban agglomeration. Finally, the urban vulnerability effect has complex feedback mechanisms and hysteresis effects, which will bring certain uncertainties to the results and require more follow-up research.

Response to Reviewer 3 Comments on “The vulnerability assessment and obstacle factor analysis of urban agglomeration along the Yellow River in China from the perspective of production-living-ecological space” 

First of all, we would like to thank you for your valuable comments. All the suggestions are of great importance for us to improve the manuscript quality. Therefore, we have revised our manuscript according to the suggestions and made point-to-point responses to the reviewer’s comments, and correcting the writing language and grammar in the new manuscript.

Point 1: The reasons for the selected indicators in the indicator system should be explained.

Response 1: Thank you for your suggestions. In the manuscript, we have detailed the reasons for the selection of indicators and the references. In addition, the optimization and innovation points of the indicator system are also explained.

Here are the added sentences.

Line205-240: From the perspective of production-living-ecological space, Thie article builds a comprehensive evaluation index system based on the concepts of representativeness, comprehensiveness, scientificity, rationality, and operability, considering the findings of previous research. Every detail of the Yellow River urban agglomeration's actual situation is taken into account. Three views are used in the construction of the comprehensive assessment index system: the production space quality, the living space quality, and the ecological space quality. In terms of the selection of production space quality and living space quality indicators, Zhang et al. 's research on production-life-ecological function of urban agglomeration in the middle reaches of the Yangtze River is mainly referred to [29]. For the selection of indicators for ecological space quality, Fang et al.'s research on the spatial differentiation of urban fragility in China is primarily consulted [30]. Although existing research has established a vulnerability evaluation index system, there are still some shortcomings that need to be addressed. For instance, existing indicators lack comprehensiveness when measuring agricultural production, as they only evaluate the overall output of agriculture without considering agricultural machinery power. Similarly, when measuring non-agricultural production, they only take industrial production into account, while ignoring the tertiary industry. Moreover, indicators for measuring the quality of living services only consider economic income and basic security, without taking into account urban population pressure, a crucial factor affecting citizens’ lives. Lastly, the indicators for measuring the quality of living space do not keep up with the times, as they do not include the important impact of tourism and leisure on residents’ quality of life. 

Therefore, this study proposes four improvements. Firstly, to more accurately evaluate agricultural production capacity, new indicators are added, such as Per unit of land gross power of agricultural machinery. Secondly, to fully assess non-agricultural production capability, per unit of land output of secondary and tertiary industries are included. Thirdly, a new dimension related to tourism and leisure is added in the measurement of living space quality. This is measured using indicators such as per unit of land domestic tourist arrivals and per unit of land domestic tourism revenue, providing a more accurate representation of residents’ quality of life in the Yellow River urban agglomeration. Finally, minor adjustments are made to the wording of indicators for ecological space quality based on the specific characteristics of the Yellow River urban agglomeration. 

In conclusion, a vulnerability assessment index system for the Yellow River urban agglomeration was constructed, including 9 first-level indicators and 27 second-level indicators (as shown in Table 2).

Point 2: The discussion section should clearly indicate the marginal contribution of this article and compare it with existing research results.

Response 2: Thank you very much and agree with your question. In the manuscript, we have added the marginal contributions of this paper, including theoretical and practical aspects, as follows:

In summary, the marginal contributions of this article are as follows: This article improves the existing indicator system and conducts a scientific analysis of the overall vulnerability and obstacle factor evolution of the urban agglomeration along the Yellow River. The high-quality development level of urban agglomerations along the Yellow River is studied not only from economic and environmental aspects, but also from social aspects. It was found that the tertiary industry, specifically tourism and leisure functions, is particularly important to urban vulnerability. Theoretically, this article enriches previous research on the Yellow River Basin and provides a new perspective on tourism and leisure research. In practice, the research results can guide the planning and management of urban agglomerations along the Yellow River, and provide reference for the sustainable development of urban agglomerations along the Yellow River in the future.

Point 3: Language expression needs improvement.

Response 3: Thank you for your suggestions. We have revised the full text language and asked two people with English writing experience to perfect it.

In addition, we have revised the figures and tables in this article. We have removed maps that did not meet the requirements of PLOS ONE and added Table 4 to show the findings more clearly. We have also optimized the language and layout of the article to meet PLOS ONE's style. After discussion and agreement among all authors, we adjusted the order of authors, that is, the first author and corresponding author were changed.

That’s all the responses.

We appreciate your suggestions sincerely.

Your efforts make our manuscript better.

---

## [Decision Letter · Decision Letter 1]

16 Jan 2024

PONE-D-23-34145R1The vulnerability assessment and obstacle factor analysis of urban agglomeration along the Yellow River in China from the perspective of production-living-ecological spacePLOS ONE

Dear Dr. Wang%,

Thank you for submitting your manuscript to PLOS ONE. After careful consideration, we feel that it has merit but does not fully meet PLOS ONE’s publication criteria as it currently stands. Therefore, we invite you to submit a revised version of the manuscript that addresses the points raised during the review process.

**Reviewer 1 **
**Comments:**

My detailed comments are as follows：

1. The font format in Figure 1 needs to be modified, and please carefully review the entire article format.

2. Please add the innovations and contributions of this paper in the literature review section to reflect the value of this paper.

3. In the first paragraph of 4.2.2, " In 2018, among the 32 provincial-level administrative regions in China, Anhui Province ranked 22nd in terms of per capita GDP, making it an economically underdeveloped province. However, in recent years, the GDP growth rate of Anhui Province has been relatively fast, and the total GDP has jumped to the tenth place in the country in 2022." The two economic concepts of per capita GDP and total GDP are used to explain the enhancement of Anhui Province's economic strength, which I think are not comparable, and it is suggested to unify the concepts.

4. In the policy recommendations, there are no specific suggestions based on the results of the paper.

5. It is suggested to add the discussion part to make the structure of the article more complete.

**Reviewer 3 **
**Comments:**

General Overview: The revised manuscript has shown considerable improvement in addressing the concerns raised in the initial review. The authors have commendably expanded their literature review, provided a more robust theoretical grounding for their methodologies, and offered a deeper analysis in the results and discussion sections.

Specific Comments:

1.Introduction: The expanded literature review now effectively situates the study within the current research landscape. The clear articulation of novel contributions significantly enhances the manuscript's relevance and academic rigor.

2.Materials and Methods: The additional theoretical justification for the choice of vulnerability measurement and barrier factor diagnosis models is appreciated. The detailed exposition of the entropy method provides clarity and strengthens the methodological framework of the study.

3.Results: The more in-depth analysis of barrier factors, with an emphasis on underlying economic theories, is a significant improvement. This approach enhances the scientific validity of the findings and their interpretation.

4.Discussion: The nuanced discussion on policy implications tailored to specific urban agglomerations is a commendable addition. These actionable recommendations considerably increase the manuscript's practical value for policymakers and practitioners.

5.Conclusion: The acknowledgment of the study's limitations and the discussion of future improvements demonstrate the authors' critical engagement with their research. The detailed policy suggestions add depth to the study's conclusions.

Recommendation: Based on the substantial improvements and the depth of revisions made, I recommend the manuscript for publication. The authors have effectively addressed the initial concerns, and the manuscript now makes a significant contribution to the field.

Suggestions for Further Improvement: Although the manuscript is considerably improved, continuous engagement with emerging research and methodologies in future work would further enhance its impact and relevance.

This review aims to provide a balanced and comprehensive assessment of the revised manuscript, recognizing the improvements made while suggesting avenues for ongoing development in the field.

We look forward to receiving your revised manuscript.

Kind regards,

Fuyou Guo, (Ph.D.

Academic Editor

PLOS ONE

Additional Editor Comments:

Reviewer 1 Comments:

My detailed comments are as follows：

1. The font format in Figure 1 needs to be modified, and please carefully review the entire article format.

2. Please add the innovations and contributions of this paper in the literature review section to reflect the value of this paper.

3. In the first paragraph of 4.2.2, " In 2018, among the 32 provincial-level administrative regions in China, Anhui Province ranked 22nd in terms of per capita GDP, making it an economically underdeveloped province. However, in recent years, the GDP growth rate of Anhui Province has been relatively fast, and the total GDP has jumped to the tenth place in the country in 2022." The two economic concepts of per capita GDP and total GDP are used to explain the enhancement of Anhui Province's economic strength, which I think are not comparable, and it is suggested to unify the concepts.

4. In the policy recommendations, there are no specific suggestions based on the results of the paper.

5. It is suggested to add the discussion part to make the structure of the article more complete.

Reviewer 3 Comments:

General Overview: The revised manuscript has shown considerable improvement in addressing the concerns raised in the initial review. The authors have commendably expanded their literature review, provided a more robust theoretical grounding for their methodologies, and offered a deeper analysis in the results and discussion sections.

Specific Comments:

1.Introduction: The expanded literature review now effectively situates the study within the current research landscape. The clear articulation of novel contributions significantly enhances the manuscript's relevance and academic rigor.

2.Materials and Methods: The additional theoretical justification for the choice of vulnerability measurement and barrier factor diagnosis models is appreciated. The detailed exposition of the entropy method provides clarity and strengthens the methodological framework of the study.

3.Results: The more in-depth analysis of barrier factors, with an emphasis on underlying economic theories, is a significant improvement. This approach enhances the scientific validity of the findings and their interpretation.

4.Discussion: The nuanced discussion on policy implications tailored to specific urban agglomerations is a commendable addition. These actionable recommendations considerably increase the manuscript's practical value for policymakers and practitioners.

5.Conclusion: The acknowledgment of the study's limitations and the discussion of future improvements demonstrate the authors' critical engagement with their research. The detailed policy suggestions add depth to the study's conclusions.

Recommendation: Based on the substantial improvements and the depth of revisions made, I recommend the manuscript for publication. The authors have effectively addressed the initial concerns, and the manuscript now makes a significant contribution to the field.

Suggestions for Further Improvement: Although the manuscript is considerably improved, continuous engagement with emerging research and methodologies in future work would further enhance its impact and relevance.

This review aims to provide a balanced and comprehensive assessment of the revised manuscript, recognizing the improvements made while suggesting avenues for ongoing development in the field.

Reviewers' comments:

Reviewer's Responses to Questions

**Comments to the Author**

1. If the authors have adequately addressed your comments raised in a previous round of review and you feel that this manuscript is now acceptable for publication, you may indicate that here to bypass the “Comments to the Author” section, enter your conflict of interest statement in the “Confidential to Editor” section, and submit your "Accept" recommendation.

Reviewer #1: (No Response)

Reviewer #2: All comments have been addressed

Reviewer #3: All comments have been addressed

2. Is the manuscript technically sound, and do the data support the conclusions?

Reviewer #1: (No Response)

Reviewer #2: Yes

Reviewer #3: Yes

3. Has the statistical analysis been performed appropriately and rigorously? 

Reviewer #1: (No Response)

Reviewer #2: Yes

Reviewer #3: Yes

4. Have the authors made all data underlying the findings in their manuscript fully available?

Reviewer #1: (No Response)

Reviewer #2: Yes

Reviewer #3: Yes

5. Is the manuscript presented in an intelligible fashion and written in standard English?

Reviewer #1: (No Response)

Reviewer #2: Yes

Reviewer #3: Yes

6. Review Comments to the Author

Reviewer #1: My detailed comments are as follows：

1. The font format in Figure 1 needs to be modified, and please carefully review the entire article format.

2. Please add the innovations and contributions of this paper in the literature review section to reflect the value of this paper.

3. In the first paragraph of 4.2.2, " In 2018, among the 32 provincial-level administrative regions in China, Anhui Province ranked 22nd in terms of per capita GDP, making it an economically underdeveloped province. However, in recent years, the GDP growth rate of Anhui Province has been relatively fast, and the total GDP has jumped to the tenth place in the country in 2022." The two economic concepts of per capita GDP and total GDP are used to explain the enhancement of Anhui Province's economic strength, which I think are not comparable, and it is suggested to unify the concepts.

4. In the policy recommendations, there are no specific suggestions based on the results of the paper.

5. It is suggested to add the discussion part to make the structure of the article more complete.

Reviewer #2: (No Response)

Reviewer #3: General Overview: The revised manuscript has shown considerable improvement in addressing the concerns raised in the initial review. The authors have commendably expanded their literature review, provided a more robust theoretical grounding for their methodologies, and offered a deeper analysis in the results and discussion sections.

Specific Comments:

1.Introduction: The expanded literature review now effectively situates the study within the current research landscape. The clear articulation of novel contributions significantly enhances the manuscript's relevance and academic rigor.

2.Materials and Methods: The additional theoretical justification for the choice of vulnerability measurement and barrier factor diagnosis models is appreciated. The detailed exposition of the entropy method provides clarity and strengthens the methodological framework of the study.

3.Results: The more in-depth analysis of barrier factors, with an emphasis on underlying economic theories, is a significant improvement. This approach enhances the scientific validity of the findings and their interpretation.

4.Discussion: The nuanced discussion on policy implications tailored to specific urban agglomerations is a commendable addition. These actionable recommendations considerably increase the manuscript's practical value for policymakers and practitioners.

5.Conclusion: The acknowledgment of the study's limitations and the discussion of future improvements demonstrate the authors' critical engagement with their research. The detailed policy suggestions add depth to the study's conclusions.

Recommendation: Based on the substantial improvements and the depth of revisions made, I recommend the manuscript for publication. The authors have effectively addressed the initial concerns, and the manuscript now makes a significant contribution to the field.

Suggestions for Further Improvement: Although the manuscript is considerably improved, continuous engagement with emerging research and methodologies in future work would further enhance its impact and relevance.

This review aims to provide a balanced and comprehensive assessment of the revised manuscript, recognizing the improvements made while suggesting avenues for ongoing development in the field.

7. PLOS authors have the option to publish the peer review history of their article (what does this mean?). If published, this will include your full peer review and any attached files.

Reviewer #1: No

Reviewer #2: No

Reviewer #3: No

---

## [Author Response · Author response to Decision Letter 1]

20 Jan 2024

Response to Reviewer 1 Comments on “The vulnerability assessment and obstacle factor analysis of urban agglomeration along the Yellow River in China from the perspective of production-living-ecological space” 

First of all, we would like to thank you for your valuable comments. All the suggestions are of great importance for us to improve the manuscript quality. Therefore, we have revised our manuscript according to the suggestions and made point-to-point responses to the reviewer’s comments, and correcting the writing language and grammar in the new manuscript.

Point 1: The font format in Figure 1 needs to be modified, and please carefully review the entire article format.

Response 1: Thank you very much and agree with your question. We have modified the font and size of Figure 1 and carefully checked the format of the full text to meet the requirements of the Plosone journal.

Point 2: Please add the innovations and contributions of this paper in the literature review section to reflect the value of this paper.

Response 2: Thank you very much and agree with your question. We have clarified the innovations and academic contributions of this article, as follows:

Line111-127: The innovations and contributions of this article are: (1) Taking the seven urban agglomerations along the Yellow River as the research object, a better understanding of the integrity and systematicity of the Yellow River urban agglomerations will help promote integration and coordination among cities. The research results have certain reference significance for urban management planning and sustainable development in other similar regions around the world. (2) On the basis of existing research, we establish an index system for evaluating vulnerability that is more comprehensive and efficiently evaluates the vulnerability of the urban agglomerations along the Yellow River between 2001 and 2020. From the perspective of production-living-ecological space, the original assessment index system is synthesized, and new indicators are creatively introduced in conjunction with the real conditions of the research region. (3) We analyze the vulnerability assessment results from the perspective of the overall vulnerability and spatial variability of the urban agglomeration. Furthermore, the obstacle factor model was used to identify the influencing factors, and then the evolution trend of the obstacle factors was analyzed. Based on these, targeted management measures are proposed for the urban agglomeration along the Yellow River.

Point 3: In the first paragraph of 4.2.2, " In 2018, among the 32 provincial-level administrative regions in China, Anhui Province ranked 22nd in terms of per capita GDP, making it an economically underdeveloped province. However, in recent years, the GDP growth rate of Anhui Province has been relatively fast, and the total GDP has jumped to the tenth place in the country in 2022." The two economic concepts of per capita GDP and total GDP are used to explain the enhancement of Anhui Province's economic strength, which I think are not comparable, and it is suggested to unify the concepts.

Response 3: Thank you for your suggestions. We have carefully checked the full text and found no issues you mentioned. The research area of this article does not involve Anhui Province. At the same time, in the comprehensive evaluation index system for the vulnerability of the urban agglomeration along the Yellow River constructed in this article, we measure the quality of living space through per capita GDP, and measure the quality of ecological space through energy consumption per 10,000 yuan of GDP, industrial wastewater discharge, and carbon emissions. There is no comparison between per capita GDP and total GDP as you pointed out. Therefore, we made no modifications here.

Point 4: In the policy recommendations, there are no specific suggestions based on the results of the paper.

Response 4：Thank you very much and agree with your question. In the manuscripts submitted earlier, we have put forward specific suggestions for each urban agglomeration for the results studied in this article, as follows:

Line733-788: Based on the above conclusions, we propose several suggestions to enhance the quality of urban development along the Yellow River urban agglomeration.

(1) Formulate a three-level development strategy. Based on the vulnerability of each urban agglomeration, local governments should formulate urban agglomeration development plans based on local conditions. Highly vulnerable urban agglomerations should implement a green circular economy model, moderately vulnerable cities need to focus on optimizing industrial structure and upgrading production technology, and low-vulnerability urban agglomerations should accelerate opening up and industrial regional cooperation. In particular, the urban agglomeration of Shandong Peninsula ought to initiate ecological restoration initiatives and cultivate a varied energy framework. The creation of a circular economy model and bolstering the growth of international commerce scale should be the priorities of the Central Plains urban agglomeration. The metropolitan agglomerations of Yu, Baotou, and Hubei should increase their investments in research and technology, introduce more professional talents, and support the modernization of production methods and the optimization of the industrial structure. The urban agglomerations of Jinzhong, Guanzhong Plain, Lanxi, and Ningxia along the Yellow River are expected to foster coordinated development and strengthen regional collaboration.

(2) Pay attention to ecological and environmental protection. Local governments should coordinate and reconcile the contradiction between ecology and development and adhere to sustainable development. For central cities with high vulnerability, some functions of the city should be distributed to surrounding cities to alleviate the pressure on the urban ecological environment; while surrounding cities should strengthen urban greening construction and promote regional cooperation to form an urban ecosystem and realize inter-city collaborative development. At the same time, local governments should establish ecological restoration projects. For upstream urban agglomerations such as Ningxia urban agglomeration, the construction of green ecological corridors should be accelerated to ensure the integrity and stability of the ecosystem in the Yellow River Basin. For urban agglomerations along the lower reaches of the Yellow River such as the Central Plains urban agglomeration, it is necessary to Continue to promote pollution prevention and control, carbon reduction and emission reduction, and enhance the ecological barrier function of the Yellow River Basin.

(3) Develop domestic and foreign trade. Due to locational constraints, the level of opening-up of urban agglomerations along the Yellow River is relatively effective. In this regard, governments at all levels should increase infrastructure investment and construction, establish a high-quality transportation network throughout the Yellow River Basin, and promote urban interconnection. At the same time, we should take Shandong Peninsula as the window of regional foreign trade, and vigorously develop export-oriented industries and carry out import and export business. And provide preferential policies to attract regional investment, improve the level of foreign trade development. 

(4) Properly plan and manage the tourism industry. Regional governments should give full play to the leading role of the tourism industry. Based on the regional Yellow River cultural advantages, relevant policies have been introduced to develop tourism into a regional advantageous industry. Urban agglomerations with high tourism and leisure vulnerability should optimize tourism resource management, enhance tourism resource utilization efficiency, and develop low-carbon tourism products and projects to reduce energy consumption in the tourism industry. For urban agglomerations with a low level of tourism industry development, they should promote technological upgrading and structural optimization and upgrading of the tourism industry by accelerating the introduction of tourism professionals and promoting talent training and information exchange. At the same time, each urban agglomeration should also vigorously promote urban tourism cooperation. This is the main way to promote the complementary advantages of the tourism industry in the upper, middle and lower reaches of the Yellow River Basin and narrow regional differences. The ultimate goal is to promote coordinated regional development.

Point 5: It is suggested to add the discussion part to make the structure of the article more complete.

Response 5：In the previously submitted manuscript, we have added a discussion section to compare this article with previous research to highlight the scientific nature and marginal contribution of this research, as follows:

Line637-704: Based on the above research results, we compare this article with previous research results to illustrate the scientific nature and marginal contribution of this article. 

Overall, the urban agglomeration along the Yellow River has a remarkable three-level gradient differential in terms of vulnerability. This characteristic is the same as earlier research, and closely corresponds to the overall vulnerability gradient differentiation of Chinese urban agglomerations [13]. This demonstrates that, as a whole, the degree of vulnerability of the urban agglomeration along the Yellow River examined in this research is congruent with the real circumstances in China. Because the gradient of vulnerability at all levels is generally consistent with previous research on urban areas in the upper, middle, and lower ranges of the Yellow River, it suggests that the study is scientific. On the other hand, the total vulnerability of the urban agglomeration along the Yellow River indicates an increasing trend, based on the specific value of the obstacle degree. The percentage of cities with high susceptibility rises, while the percentage of cities with moderate or high vulnerability stays constant. This is consistent with previous research conclusions on urban agglomerations in the Yangtze River Basin [34,35], Pearl River Basin [36] and other regions, but is contrary to research conclusions in northeast China [37], central Yunnan [38], Zhangjiakou [39] and other regions of China. It is evident that the vulnerability of the urban agglomeration along the Yellow River has a particular influencing mechanism.

Urban agglomeration is a coupled system of economy, society and environment. Prior research on the Yellow River Basin has mostly concentrated on environmental and economic issues [40-43]. The findings indicate that the primary determinants of the urban agglomeration's growth along the Yellow River are resource availability, ecological environment, and economic development. The research conclusions of this article have a high degree of fit with it. However, existing studies lack the perspective of living space quality. Although there are studies on social and people's livelihood aspects, they are more focused on policy support and lack research on urban tourism and leisure functions [44,45]. It can be seen that the existing research on the quality of urban development in the Yellow River Basin is incomplete. Therefore, this paper constructs a multi-dimensional index system to comprehensively evaluate the evolutionary characteristics of vulnerability barrier factors in the urban agglomeration along the Yellow River. Compared with existing evaluation methods, this index system is more systematic and scientific.

Contrary to earlier research, the study's findings indicate that the influence of leisure and tourism activities on the vulnerability of urban agglomerations along the Yellow River has grown dramatically and is especially significant. We proceed from an economic perspective based on supply and demand effects. In terms of demand, with the continuous development of China's economy and the continuous increase in national income, the demand for tourism is becoming increasingly strong. This has led to a continuous increase in the number of tourists per region, which has put forward higher-level requirements for the development of tourism in urban agglomerations along the Yellow River; At the same time, as China's economy shifts from high-speed development to high-quality development, the country has attached great importance to the protection of the Yellow River culture and the development of the tourism industry since 2020. Higher requirements have been placed on the average tourism income of urban agglomerations along the Yellow River, which has led to the increasingly prominent importance of tourism and leisure functions in the development of urban vulnerability. In terms of supply, urban agglomerations along the Yellow River mainly focused on the development of primary and tertiary industries in the past, with the tertiary industry accounting for a relatively low proportion. The tourism supply was seriously insufficient and difficult to meet social needs. Although the industrial structure has improved in recent years and the tertiary industry has developed, the development level of the tourism industry is not high, mostly at the expense of consuming resources and damaging the environment. The average number of tourists per area and the average tourism income are not high, which is extremely detrimental to the high-quality development of the city and increases the vulnerability of the urban agglomeration along the Yellow River.

In summary, the marginal contributions of this article are as follows: This article improves the existing indicator system and conducts a scientific analysis of the overall vulnerability and obstacle factor evolution of the urban agglomeration along the Yellow River. The high-quality development level of urban agglomerations along the Yellow River is studied not only from economic and environmental aspects, but also from social aspects. It was found that the tertiary industry, specifically tourism and leisure functions, is particularly important to urban vulnerability. Theoretically, this article enriches previous research on the Yellow River Basin and provides a new perspective on tourism and leisure research. In practice, the research results can guide the planning and management of urban agglomerations along the Yellow River, and provide reference for their sustainable development.

That’s all the responses.

We appreciate your suggestions sincerely.

Your efforts make our manuscript better.

---

## [Decision Letter · Decision Letter 2]

15 Feb 2024

The vulnerability assessment and obstacle factor analysis of urban agglomeration along the Yellow River in China from the perspective of production-living-ecological space

PONE-D-23-34145R2

Dear Dr. Wang,

We’re pleased to inform you that your manuscript has been judged scientifically suitable for publication and will be formally accepted for publication once it meets all outstanding technical requirements.

Kind regards,

Fuyou Guo, (Ph.D.

Academic Editor

PLOS ONE

Additional Editor Comments (optional):

Reviewers' comments:

Reviewer's Responses to Questions

**Comments to the Author**

1. If the authors have adequately addressed your comments raised in a previous round of review and you feel that this manuscript is now acceptable for publication, you may indicate that here to bypass the “Comments to the Author” section, enter your conflict of interest statement in the “Confidential to Editor” section, and submit your "Accept" recommendation.

Reviewer #2: All comments have been addressed

2. Is the manuscript technically sound, and do the data support the conclusions?

Reviewer #2: Yes

3. Has the statistical analysis been performed appropriately and rigorously? 

Reviewer #2: Yes

4. Have the authors made all data underlying the findings in their manuscript fully available?

Reviewer #2: Yes

5. Is the manuscript presented in an intelligible fashion and written in standard English?

Reviewer #2: Yes

6. Review Comments to the Author

Reviewer #2: This article has been revised in response to my previous comments, and I believe it meets the requirements for publication.

7. PLOS authors have the option to publish the peer review history of their article (what does this mean?). If published, this will include your full peer review and any attached files.

Reviewer #2: No

---

## [Editor Report · Acceptance letter]

23 Mar 2024

PONE-D-23-34145R2 

PLOS ONE

Dear Dr. Wang, 

I'm pleased to inform you that your manuscript has been deemed suitable for publication in PLOS ONE. Congratulations! Your manuscript is now being handed over to our production team.

Kind regards, 

on behalf of

Associate professor Fuyou Guo 

Academic Editor

PLOS ONE